# Oncolytic Therapies for Glioblastoma: Advances, Challenges, and Future Perspectives

**DOI:** 10.3390/cancers17152550

**Published:** 2025-08-01

**Authors:** Omar Alomari, Habiba Eyvazova, Beyzanur Güney, Rana Al Juhmani, Hatice Odabasi, Lubna Al-Rawabdeh, Muhammed Edib Mokresh, Ufuk Erginoglu, Abdullah Keles, Mustafa K. Baskaya

**Affiliations:** 1Hamidiye International School of Medicine, University of Health Sciences, 34668 Istanbul, Türkiye; dromari2001@gmail.com (O.A.); hebibeyvaz@gmail.com (H.E.); beyzaguneyst@gmail.com (B.G.); m.edib.mokresh@gmail.com (M.E.M.); 2Faculty of Medicine, Baskent University, 06790 Ankara, Türkiye; ranajamal200@gmail.com; 3Department of Medical Oncology, Kartal Dr. Lütfi Kirdar City Hospital, Health Science University, 34865 Istanbul, Turkey; odabashatice@yahoo.com; 4School of Medicine, Yarmouk University, Irbid 21163, Jordan; lobna.alrawabdeh2001@gmail.com; 5Department of Neurosurgery, University of Wisconsin School of Medicine, Madison, WI 53706, USA; erginoglu@wisc.edu (U.E.); abdullah.keles@wisc.edu (A.K.)

**Keywords:** blood-brain barrier, immunotherapy, GBM, Oncoviral

## Abstract

Glioblastoma (GBM) is a highly aggressive brain tumor with limited treatment options, prompting interest in oncolytic therapies, particularly oncolytic viruses (OVs), that can destroy tumor cells and activate immune responses. Promising viral candidates include modified herpes simplex virus, adenovirus, poliovirus, and Newcastle disease virus, each offering tumor-selective advantages. While some OV therapies like G207 and Delytact™ have shown clinical potential, challenges such as immune resistance, delivery obstacles, and patient variability remain. Combining OVs with other treatments, such as immune checkpoint inhibitors or chemotherapy, offers potential but needs further validation. Additionally, non-viral oncolytic agents, like tumor-targeting bacteria, have been emerging, yet they are underexplored. To move forward, research must focus on improving OV delivery, tailoring treatments using biomarkers, and addressing the immunosuppressive tumor environment.

## 1. Introduction

Glioblastoma (GBM) is the most frequent primary malignant brain tumor in adults, representing nearly half of all primary central nervous system (CNS) cancers. Its estimated annual incidence is around 3 cases per 100,000 individuals [1]. GBMs usually arise de novo and are characterized by aggressive local invasion into surrounding brain tissue, though they rarely metastasize to other organs [2]. To date, the only well-established GBM risk factor is exposure of the CNS to ionizing radiation [3]. The current standard of care includes maximal safe surgical resection, followed by radiotherapy in combination with concurrent and adjuvant temozolomide chemotherapy [4]. Despite current treatments, the prognosis for GBM patients remains extremely poor, with fewer than 3% surviving five years post-diagnosis, with older age being the most consistent predictor of unfavorable outcomes [5].

Vascular endothelial growth factor (VEGF) is markedly overexpressed in GBM, driving abnormal neovascularization and aggressive tumor growth. Consequently, anti-VEGF therapies have emerged as important adjuncts in GBM management by targeting the tumor’s angiogenic dependency [6]. Among these, bevacizumab received accelerated approval from the FDA in 2009 for the treatment of recurrent GBM following prior therapy, supported by phase II trial data demonstrating meaningful clinical benefits, including improved 6-month progression-free survival, enhanced radiographic and neurologic responses, and reduced cerebral edema, despite no significant improvement in overall survival [7]. Additionally, in one cohort, bevacizumab significantly reduced corticosteroid requirements in patients receiving steroids at baseline, with 83.3% achieving more than a 50% dose reduction, reflecting improved peritumoral edema control and contributing to better quality of life [8]. Overall, bevacizumab remains a valuable adjunct in GBM management for improving disease control and symptom burden, despite the absence of an overall survival advantage.

Even with first-line therapies, GBM almost invariably recurs, with median overall survival after recurrence ranging from 2 to 9 months [9]. This dismal outlook is largely attributable to several challenges, including significant intertumoral and intratumoral heterogeneity that promotes the emergence of resistant cell populations; tumors in difficult anatomical locations shielded by the blood–brain barrier (BBB), thus limiting effective drug delivery; and an immunosuppressive tumor microenvironment [10]. Additionally, GBM is classified as a “cold tumor” because of its immunosuppressive environment and weak T-cell response. As a result, immune checkpoint immunotherapy and chimeric antigen receptor (CAR) T-cell therapy have shown limited success for GBM treatment [5,10].

Traditional treatments are unable to surmount these obstacles, underscoring the urgent need for innovative therapies [11]. Oncolytic virotherapy (OVT) represents a promising new personalized approach for addressing these GBM treatment challenges.

Viruses and bacteria, once viewed as harmful, are now recognized as powerful allies for GBM therapy. Their diversity, selectivity, and manipulability enable them to replicate within to destroy tumor cells while leaving normal cells undamaged, achieving what conventional treatments often fail to accomplish [4,12]. Oncolytic viruses (OVs) have been the focus of many experimental GMB studies due to their compatibility with the brain’s unique microenvironment, the absence of distant metastasis with GBM, and the tumor’s rapid cell division, which facilitates viral replication [10,13,14,15,16] (Figure 1).

OVs exert antitumor effects through both direct cytolytic mechanisms and immune-mediated responses. Direct oncolysis involves the selective replication of viruses within tumor cells, leading to cell rupture and local inflammation [17]. More critically, OVs initiate a robust antitumor immune response by releasing damage-associated molecular patterns (DAMPs) and pathogen-associated molecular patterns (PAMPs) during tumor cell lysis [18,19]. These molecular signals stimulate the production of type I interferons (IFNs) and other proinflammatory cytokines, activating dendritic cells, macrophages, and other antigen-presenting cells (APCs), which in turn prime tumor- and virus-specific CD4^+^ and CD8^+^ T cells [19]. This dual stimulation of innate and adaptive immunity is believed to overcome the immunologic tolerance that characterizes GBM, effectively transforming it from a “cold” to a “hot” tumor [19]. This pleiotropic immune activation has been associated with improved survival in subsets of GBM patients.

These mechanisms highlight the novel and promising potential of OVT for GBM treatment by effectively addressing many of the limitations of traditional therapies. By overcoming obstacles including the tumor’s complex anatomical location, extensive heterogeneity, BBB infiltration, microscopic invasion, and immunosuppressive microenvironment, OVT offers a new therapeutic avenue [10].

Encouraging outcomes from both preclinical and clinical studies, most notably the conditional approval of the oncolytic virus G47∆ (Delytact) in Japan for brain tumors, further reinforce the promise of this approach, making it more acceptable among physicians [16]. This growing success of OVT marks a potentially game-changing development for GBM treatment, bringing renewed optimism in the fight against this highly aggressive and lethal disease.

This review summarizes the current landscape of oncolytic therapies for glioblastoma, encompassing both viral and non-viral approaches. It examines the underlying mechanisms and clinical progress of key oncolytic viruses, including Herpes Simplex Virus [HSV], Adenovirus, Poliovirus, and Reovirus. We also highlight combination therapies that are designed to enhance OVT efficacy. And finally, we explore emerging non-viral alternatives and outline future directions in the field by highlighting innovations in delivery systems, biomarker-guided patient selection, and the integration of oncolytic agents with targeted therapies.

## 2. Virus-Based Oncolytic Therapies

Multiple viral platforms, including adenovirus, HSV, reovirus, and measles virus, have been investigated for their unique mechanisms of action and tumor-targeting capabilities in preclinical GBM models. These approaches leverage both direct oncolysis and the modulation of the tumor microenvironment to reduce tumor burden and enhance radiosensitivity, providing multifaceted in vivo treatment strategies as illustrated in Figure 2.

### 2.1. Herpes Simplex Virus

Herpesviruses are double-stranded DNA viruses, with nine types known to infect humans, most notably herpes simplex virus 1 (HSV-1) and HSV-2. These viruses can establish lifelong latent infections, particularly in neural tissues, and may reactivate under immunosuppressive conditions, leading to lytic infections and associated clinical complications [20]. During lytic infection, numerous viral gene products are expressed, whereas latent infection is characterized by limited gene expression [21,22]. HSV encodes over 70 genes, and its interaction with host cells plays a crucial role in controlling both latency and reactivation. Since HSV can fuse with cell membranes and enter host cells using specific receptor-ligand interactions and selectively lyse tumor cells following tumor cell-specific virus replication, it holds strong promise for oncolytic therapy.

Herpes simplex virus type 1 presents a highly favorable candidate for oncolytic virotherapy, owing to its unique biological features and well-established clinical profile. With a large genome (~152 kb), HSV-1 contains non-essential sequences that can be safely removed to accommodate therapeutic genes without compromising viral replication. Importantly, since the viral DNA does not integrate into the host genome, the risk of insertional mutagenesis is significantly minimized [23]. Although HSV-1 is extensively characterized as a human pathogen from a clinical perspective, its use as an oncolytic virus required substantial genetic engineering to attenuate its neurovirulence and enhance tumor selectivity. These modifications have enabled HSV-1 derivatives, such as G207 and M032, to safely and effectively target malignant cells while sparing healthy tissue. The availability of approved antiviral therapies, such as acyclovir, offers a crucial safety net to manage potential off-target viral activity [24]. Preclinical mouse studies have shown that oncolytic HSV-1 selectively infects and destroys tumor cells without damaging healthy tissues. In addition to direct tumor cell lysis, this process also triggers an antitumor immune response, which promotes anti-tumor immunity (immunovirotherapy) [25]. This dual effect of HSV-1—oncolysis and immune activation—makes it a promising candidate for oncolytic virotherapy. To date, numerous viruses have been genetically modified as oncolytic agents and tested in clinical trials against various cancer types, including GBM. Among these, HSV-1-based Talimogene laherparepvec (T-VEC) remains the only oncolytic virotherapy agent approved for the treatment of advanced melanoma by both the U.S. Food and Drug Administration (FDA) and the European Medicines Agency (EMA) [15].

T-VEC is a product of the deletion of the γ34.5 virulence gene, which normally blocks the PKR pathway, and the US12 gene, which inhibits antigen presentation in infected cells. HSV-1 preserves protein synthesis during infection by preventing the PKR-mediated phosphorylation of eIF2α through its ICP34.5 protein, thereby counteracting the host antiviral response, while ICP 47 suppresses immune response by preventing antigen presentation via MHC class I molecules [26,27,28]. Additionally, the US 11 protein counteracts host cell protein synthesis shutdown, thereby overcoming viral suppression. This simultaneous alteration of gene dynamics finely adjusts the virulence of oHSV-1, enabling it to replicate in cancer cells that are moderately resistant.

In the face of these challenges, GBM stands out as a promising candidate for oncolytic virotherapy. The neuro-attenuated oHSV1 vector is currently being tested in both human and non-human studies.

#### 2.1.1. Herpes Simplex Virus Oncolysis in Lab and Animal Studies

A broad array of genetically modified oncolytic herpes simplex viruses (oHSVs) has demonstrated promising antitumor efficacy in preclinical GBM models (Table 1) [29,30,31,32,33,34,35,36,37,38,39,40,41,42,43,44,45,46,47,48,49,50,51,52,53,54,55,56]. Variants engineered to express immune-modulating cytokines (e.g., IL-2, IL-12) or tumor suppressors (e.g., PTEN isoforms) enhanced immune cell infiltration, reversed immunosuppressive signaling pathways such as IL6/STAT3 and PI3K/AKT, and overcame resistance to radiotherapy and checkpoint inhibition. Tumor-targeted constructs, such as HER2-, EGFRvIII-, and CXCR4-specific oHSVs, showed selective cytotoxicity against glioma stem-like cells (GSCs) and improved survival, particularly in early-stage disease. Several approaches addressed the intrinsic resistance of GSCs to viral replication by targeting pathways like TNF-α, TGF-β, JNK, and STAT3 using pharmacologic inhibitors or adjunctive methods such as fasting. Combinatorial strategies, including oHSV with CAR-T or NK cells, MEK inhibitors, bortezomib, or extracellular matrix-degrading enzymes (e.g., ChaseM), yielded synergistic effects, enhancing viral spread, immune activation, and tumor regression. Additionally, modifying the tumor microenvironment (such as blocking HMGB1 to reduce edema or degrading chondroitin sulfate proteoglycans to improve penetration) was crucial for therapeutic success. These findings suggest that future efforts should focus on personalized oHSV design based on tumor profiles, rational combinations with immunotherapies or targeted agents, and innovative delivery strategies to sustain viral activity and remodel the tumor milieu. Translation into early-phase clinical trials with biomarker-driven stratification will be essential to realize the full potential of oHSV-based therapies in GBM.

#### 2.1.2. Herpes Simplex Virus Oncolysis in Clinical Studies

Recent clinical trials exploring oHSV therapy for GBM have produced encouraging early-phase results, highlighting both safety and therapeutic potential (Table 2) [57,58,59,60,61]. Various engineered HSV strains, including G207, G47∆, M032, and CAN-3110, were tested across adult and pediatric populations with relapsed or treatment-resistant GBM. These studies demonstrated that oHSV therapy is generally well tolerated, with manageable side effects and evidence of immune activation within the tumor microenvironment. In several cases, radiographic and histological improvements were observed, along with extended survival in select patients.

Importantly, these trials revealed that oHSV can convert immunologically “cold” tumors into “hot” ones by enhancing lymphocyte infiltration and triggering immune-related gene expression. Pre-existing HSV immunity also appeared to enhance therapeutic response in some patients, suggesting a possible role for personalized approaches in future applications. Despite variability in survival outcomes and adverse events, the overall safety and immunostimulatory effects of oHSV support its continued investigation.

Future trials should focus on optimizing viral engineering, improving delivery methods, and combining oHSV with treatments such as radiotherapy or immunotherapy. Larger, controlled studies are needed to confirm efficacy and guide its integration into clinical practice. Ultimately, oHSV represents a promising frontier in the treatment of GBM, with potential to overcome many limitations of current therapeutic options.

### 2.2. Adenovirus

Adenoviruses (AdVs) are non-enveloped, double-stranded DNA viruses from the Adenoviridae family, with over 100 identified serotypes, 49 of which infect humans [62,63]. These viruses are highly prevalent worldwide and can infect various organs including the respiratory tract, gastrointestinal system, kidneys, and conjunctiva [62]. While most infections are asymptomatic or mild, AdV can cause severe complications including hemorrhagic cystitis, pancreatitis, and meningoencephalitis, especially in immunocompromised individuals. Transmission occurs via respiratory droplets, direct contact, or the fecal-oral route [64]. AdVs display organ-specific tropism based on serotype, with infection severity and location often influenced by the entry route. These viruses can remain latent in tonsils, lymphocytes, and other tissues. Their 26 to 45 kbp genome contains early, intermediate, and late transcription regions responsible for replication and viral gene expression [65,66]. With an icosahedral capsid structure and significant dependence on host cells for replication, AdVs have both lytic and latent phases and possess oncogenic potential, particularly through E1A-mediated transformation in experimental models.

Adenoviruses have recently gained attention as oncolytic agents for cancer immunotherapy due to their large gene-carrying capacity and tumor-selective replication. Oncolytic adenoviruses (oAds) destroy tumor cells directly through lysis and indirectly by stimulating immune responses, without harming healthy cells [67]. They can be genetically engineered to carry immunostimulatory genes or to target tumors with specific mutations such as p53 deficiency, as seen in Onyx-015 and H101 (Oncorine), which is approved in China for head and neck cancers [68]. These engineered viruses improve T cell activation, antigen presentation, and immune modulation. In addition, oAds show promise by disrupting the immunosuppressive tumor microenvironment and triggering immune-mediated GBM tumor cell destruction. By enhancing antigen release and activating cytotoxic T cells via DAMPs and PAMPs, oAds may offer a much-needed therapeutic GBM strategy, with ongoing studies exploring their clinical potential.

#### 2.2.1. Adenovirus Oncolysis in Lab and Animal Studies

The summarized studies in Table 3 present evidence that genetically modified oAdVs hold significant promise for GBM treatment [69,70,71,72,73,74,75,76,77,78,79,80,81,82,83,84,85,86,87,88,89,90,91,92,93,94,95,96,97,98,99,100,101,102,103,104,105]. One of the most consistent findings across these models is that replication-competent oAdVs elicit stronger immune responses and better tumor control compared to non-replicating vectors. Key genetic modifications such as capsid engineering (e.g., Ad5/Ad37 fiber swaps), incorporation of immunostimulatory genes like IL-15, or the addition of tumor-targeting promoters (e.g., Ki67, hTERT) enable selective targeting of tumor cells and enhance antitumor immune activity. Some studies demonstrated that combining oAdVs with immune checkpoint inhibitors (e.g., anti-PD-1/PD-L1) or standard therapies (e.g., temozolomide or olaparib) further amplify therapeutic outcomes. Notably, carriers like mesenchymal stem cells improved viral delivery and immune evasion, and novel delivery routes such as intranasal administration increased survival in animal models, underscoring the importance of optimizing both vector design and delivery strategies.

Collectively, these findings suggest that future therapeutic directions should prioritize multi-modal strategies that integrate oAdVs with immunotherapy, targeted delivery systems, and tumor-specific promoters to enhance efficacy while minimizing off-target effects. The development of third-generation vectors (e.g., TS-2021) and exploration of underutilized serotypes (e.g., Ad6) illustrate ongoing innovations. Future studies should also explore personalized oAdV therapies based on patient-specific tumor profiles and immune landscapes.

#### 2.2.2. Adenovirus Oncolysis in Clinical Studies

Published clinical studies highlight the growing promise of oAdvs as a therapeutic GBM modality (Table 4) [106,107,108,109,110]. Early-phase trials, such as Fares et al. using NSC-CRAd-S-pk7, demonstrated not only a manageable safety profile but also a median survival of 18.4 months which is considered an encouraging outcome for newly diagnosed patients [100]. Similarly, Delta24-RGD (DNX-2401), delivered via convection-enhanced delivery, showed complete tumor regression in one patient who survived for eight years, reflecting the potential for durable responses. Importantly, the addition of RGD motifs allowed viral entry despite the presence of neutralizing antibodies, broadening the applicability of these therapies even in pre-immunized individuals. Combinatorial approaches involving immune checkpoint inhibitors like pembrolizumab have also yielded promising results, with improved median survival (12.5 months) and no dose-limiting toxicities, suggesting synergy between viral oncolysis and immunotherapy.

Despite these advances, study sample sizes were small, limiting the generalizability of findings. Median survival varied widely between trials, likely reflecting differences in patient populations (e.g., newly diagnosed vs. recurrent GBM), delivery methods, and virus design. Furthermore, while oAdv therapy has shown immune activation and tumor responses, the variability in infusion area and delivery precision noticed in current studies underscores the need for optimized delivery protocols. Future studies should focus on larger, randomized trials that compare oAdv monotherapy and combination regimens with standard-of-care treatments. Moreover, integrating molecular profiling to select patients, most likely to benefit and understand mechanisms of resistance, will be crucial for advancing oAdv-based strategies into mainstream neuro-oncology practice.

### 2.3. Measles Viruses

The measles virus (MeV), a highly contagious negative-strand RNA virus from the Paramyxoviridae family, has shown promising oncolytic potential due to its natural tropism for tumor cells, ability to induce syncytia formation, and capacity to stimulate strong virus-specific immune responses [111,112]. The attenuated Edmonston strain (MV-Edm) has been genetically engineered and demonstrated to have durable responses in preclinical GBM models as well as good safety profiles in early-phase clinical trials (Table 5) [113,114,115,116,117,118]. However, antiviral resistance mechanisms in tumor cells remain a barrier. In one recent study, they concluded that MeV encoding miR-122 may have the potential to reduce target protein levels by 40%, though Drosha-dependent nuclear retention limited miRNA production [114]. Another study using patient-derived xenograft (PDX) models identified the JAK1 pathway as a key regulator of MeV replication, and combining MeV with the JAK1 inhibitor ruxolitinib improved viral replication and treatment efficacy, highlighting the need for further integration with immune-modulating strategies such as T-cell monitoring [115].

### 2.4. Newcastle Disease Virus

Newcastle Disease Virus (NDV), an avian paramyxovirus, has shown strong potential as a GBM oncolytic agent due to its ability to selectively replicate in tumor cells [119]. NDV induces cancer cell death through intrinsic and extrinsic caspase pathways, ER stress responses, and the release of TNF-α, while triggering robust inflammatory and interferon responses via its HN and F proteins [120]. GBM cells often lack effective type I interferon signaling, making them especially vulnerable to NDV-mediated oncolysis [121]. Recombinant NDV (rNDV) strains further enhance this effect, although preexisting immunity remains a challenge. Current strategies focus on improving efficacy through molecular modifications and combination therapies [122].

The preclinical studies indicate that NDV is a promising oncolytic virotherapy agent for GBM (Table 6) [121,123,124,125,126,127,128,129]. Although more research is needed for clinical application, combining NDV with Temozolomide enhances its oncolytic effects. The interferon pathway deficiency in glioblastoma cells allows for greater viral replication, which helps suppress tumor growth. Animal studies have also shown increased efficacy of recombinant NDV-PTEN. Additionally, using an antiviral serum against interferon viruses alongside NDV may still be effective in patients with existing virus immunity. Though clinical research on NDV in GBM is limited, existing studies suggest notable potential (Table 6). Further investigation is needed to optimize dosing, delivery timing (especially before anti-NDV antibodies develop), and to identify tumor profiles most responsive to therapy. NDV-infected tumor vaccines show strong peripheral immune activation, but understanding their effect within the tumor microenvironment is crucial, particularly in overcoming GBM’s immunosuppressive mechanisms. While no active clinical trials currently explore NDV for GBM, preclinical research remains active, supporting the need for future clinical studies to evaluate its safety and therapeutic impact.

NDV is well-known as a highly contagious and pathogenic virus in avian species, raising understandable regulatory, ecological, and zoonotic safety concerns regarding its use in human clinical settings. Despite this, extensive preclinical and clinical investigations have demonstrated that attenuated and recombinant NDV strains can be safely administered to patients, with no serious adverse events reported to date (Table 6). The use of NDV oncolytic virotherapy in several clinical trials has been presented as a safety strategy for treating various solid tumors [121,130], including melanoma [131], renal cell carcinoma [132], and ovarian cancer [133], among others. These studies collectively support a favorable safety profile under controlled conditions and adherence to strict biosafety measures and dosing protocols. Furthermore, the lack of preexisting immunity to NDV in humans and the virus’s selective replication in tumor cells deficient in type I interferon signaling provide a compelling rationale for its continued development as an oncolytic agent. Nevertheless, ongoing vigilance is necessary to monitor potential environmental and regulatory risks, and careful patient selection based on tumor interferon pathway status may enhance therapeutic efficacy while minimizing safety concerns.

### 2.5. Reovirus

Reovirus (Pelareorep), a wild-type Type 3 Dearing strain, has emerged as a promising oncolytic virotherapy for GBM treatment by exploiting the dysregulated Ras signaling pathway commonly found in malignant cells [13]. In normal cells, reovirus replication is blocked by an innate antiviral response mediated by protein kinase R (PKR), which inhibits viral protein synthesis [134]. However, in GBM cells with constitutively active Ras signaling, this PKR-mediated defense is impaired, allowing efficient viral replication, selective tumor cell lysis, and sparing of healthy tissue. Beyond direct oncolysis, reovirus infection promotes immunogenic cell death, releasing tumor-associated antigens and proinflammatory cytokines that stimulate systemic antitumor immunity. This dual mechanism (targeted cytotoxicity and immune activation) positions Pelareorep as a promising candidate against GBM, especially when combined with other immunotherapies or standard treatments to enhance efficacy and overcome resistance.

The current published literature demonstrates that Pelareorep-based reovirus therapy, alone or combined with GM-CSF, is generally safe and well tolerated across diverse glioma patient populations, including recurrent malignant gliomas, newly diagnosed GBM, and pediatric high-grade gliomas (Table 7) [135,136,137,138]. Viral delivery to tumor sites was consistently confirmed, with evidence of blood–brain barrier penetration and immunogenic effects including increased T cell activity and tumor cell death. However, while early biological and immunological engagement is promising, clinical efficacy remains limited, especially for pediatric high-grade gliomas where rapid tumor progression and antiviral antibody neutralization likely diminished therapeutic benefit. These findings suggest that future research should focus on strategies to enhance sustained viral activity and overcome immune neutralization, potentially through combination therapies or immune modulation, to translate these encouraging biological effects into meaningful clinical outcomes.

### 2.6. Retroviruses

Retroviruses, particularly replication-competent retroviruses (RCRs) and retroviral replicating vectors (RRVs), have emerged as promising agents in oncolytic virotherapy for GBM due to their ability to stably integrate therapeutic genes into tumor genomes. Preclinical studies have demonstrated the efficacy of various engineered retroviral vectors such as MLV-based RCRs, Toca 511, and modified foamy viruses in delivering suicide genes like cytosine deaminase or inducible caspase systems, leading to tumor-selective replication, sustained gene expression, and enhanced survival in animal models (Table 8) [139,140,141,142]. Clinical trials, especially those investigating Toca 511 in combination with the prodrug Toca FC, have shown favorable safety profiles and encouraging survival outcomes, with some patients achieving durable complete responses [142,143,144,145,146]. Molecular and immunologic analyses suggest immune-related mechanisms contribute to therapeutic benefit, reinforcing the potential of retrovirus-based strategies in personalized cancer therapy. Despite challenges such as variability in patient response and vector behavior, the accumulated evidence supports further development of retroviral platforms for effective, tumor-selective gene therapy in GBM.

### 2.7. Vaccinia Virus

Recent advancements in vaccinia virus (VV)-based oncolytic therapies highlight promising breakthroughs for GBM treatment. Several innovative VV constructs have demonstrated strong tumor selectivity, BBB penetration, and the ability to reshape the immunosuppressive tumor microenvironment. A new generation VV armed with interleukin-21 (IL-21) synergized with anti-PD1 therapy promoted durable immune memory to achieve long-term tumor control in murine models [147]. Another construct, ΔF4LΔJ2R VV, showed enhanced efficacy when combined with radiotherapy, curing the majority of treated mice and improving CD8+/Treg ratios [148]. VV-GMCSF-Lact, engineered to express GM-CSF and lactaptin, showed cytotoxicity against patient-derived GBM cells and successful tumor targeting after intravenous delivery [149]. Additionally, a BMP-4-expressing VV demonstrated strong activity against glioma stem cells, preventing tumor recurrence and significantly improving survival in preclinical models [150]. Although another IL-12-expressing VV (VVL-m12) was developed primarily for lung cancer, it also demonstrated potent antitumor activity, strong immunomodulation, and synergy with α-PD1 therapy—findings with potential translational relevance for GBM treatment [151]. Collectively, these studies underscore the therapeutic potential of VV-based virotherapy in GBM, especially when combined with immune checkpoint inhibitors or radiation. Future research should focus on clinical translation and optimizing combinations that overcome GBM’s immunoresistance and recurrence.

### 2.8. Parvovirus H-1

Parvovirus H-1 (H-1PV), a rodent-derived protoparvovirus from the *Parvoviridae* family, exhibits strong tumor tropism due to dysregulated DNA damage responses and interferon signaling in cancer cells, while remaining non-pathogenic to normal cells [152]. Its oncolytic effects are primarily mediated by the NS1 protein, which induces replication stress, caspase-dependent apoptosis, and immunogenic cell death marked by DAMP release and subsequent immune activation [153,154]. Because productive parvovirus infection requires cell division and is enhanced by oncogenic transformation, certain parvoviruses show promise as oncolytic agents. Mechanistic studies revealed that H-1PV induces apoptosis in tumor cells by activating the CPP32 ICE-like cysteine protease, leading to PARP cleavage and TNF-alpha-like morphological changes, with selective toxicity toward tumor cells over normal astrocytes [155,156]. In vitro study aimed to screen of 12 parvoviruses against human glioblastomas, LuIII, H-1, MVMp, and MVM-G52 showed the highest oncolytic activity at high multiplicity of infection [157]. At low multiplicity of infection, only LuIII effectively infected and killed all five glioblastoma cell lines tested, while H-1 was effective in two [157]. LuIII’s enhanced oncolysis was linked to its capsid and full genome, which enabled selective infection of glioma cells over normal glia in vitro and in vivo [157]. In the used mouse models at the same study, LuIII significantly reduced tumor growth without adverse effects, even after intravenous or intracranial administration [157]. In a Phase I/IIa clinical trial (NCT01301430) involving 18 patients with progressive or recurrent GBM, ParvOryx was administered either intratumorally or intravenously at various doses [158]. The treatment was well tolerated with no dose-limiting toxicities and achieved a median overall survival of 15.5 months [158,159]. Viral transcripts were detected in tumors regardless of dose or administration route, indicating that ParvOryx can cross the blood–brain barrier (BBB) [158]. Moreover, treatment enhanced tumor infiltration by CD4^+^ and CD8^+^ T cells and increased pro-inflammatory cytokines in six patients [159]. Despite these promising results, no further clinical trials in GBM are currently underway.

### 2.9. Poliovirus

Poliovirus recombinant Sabin-rhinovirus IRES (PVSRIPO) is an engineered oncolytic poliovirus derived from the Sabin type 1 strain, modified with a human rhinovirus type 2 IRES to reduce neurovirulence while preserving its ability to target tumors overexpressing CD155, a receptor highly expressed in GBM [160]. In GBM, *CD155* found to be consistently expressed across a wide range of models, including laboratory and primary GBM cell lines, patient-derived explant cells, GBM biopsy-derived xenografts, and surgically resected tumor tissues [161,162], underscoring its relevance as a potential therapeutic target in this aggressive brain tumor. PVSRIPO replicates in CD155^+^ tumor and APCs, leading to direct oncolysis and innate immune activation through MDA-mediated type I interferon signaling, which enhances dendritic cell maturation and CD8^+^ T cell priming [163,164]. 

In vivo, PVSRIPO reduced tumor burden and prolonged survival in murine GBM models, with systemic immune responses and no signs of neurovirulence in primate studies [165,166]. Clinically, a notable result from the Phase I trial (NCT01491893) reported a complete and durable radiologic remission in a patient with recurrent GBM, who remained disease-free for 57 months following treatment—one of the longest survivals reported with virotherapy in GBM [166]. Moreover, combining PVSRIPO with anti–PD-1 therapy may lead to enhanced T cell infiltration and better outcomes as reported in other malignancies such as melanoma [167]. 

### 2.10. Zika Virus

Zika virus (ZIKV), a neurotropic, mosquito-borne Flavivirus, demonstrates promising oncolytic potential against GBM, particularly by targeting glioma stem cells (GSCs), which are critical drivers of tumor growth and recurrence [168,169]. ZIKV exploits the overexpression of AXL and its ligand Gas6 in GBM cells for cellular entry, preferentially infecting and killing GSCs while sparing differentiated glioma and normal neural cells [170,171]. Its selective cytotoxicity is mediated through factors like elevated SOX2 and integrin αvβ5, alongside suppressed innate immune responses in GSCs [172]. In vitro studies show ZIKV induces rapid cytopathic effects and apoptosis in GBM cells, possibly through digoxin elevation and TNF-α upregulation, with NS5 identified as the primary inhibitory viral protein [173,174,175]. In vivo, ZIKV monotherapy improved survival by targeting GSCs and eliciting robust CD8^+^ T cell responses [176], with combination therapies such as with anti–PD-1 or irradiation, yielding synergistic benefits [177]. Although no clinical trials have tested ZIKV in GBM, a compelling case report described a GBM patient experiencing long-term remission following natural ZIKV infection, suggesting a potential real-world therapeutic effect worth further investigation [178].

## 3. Combination Oncolytic Therapies

There is no doubt that OVT offers powerful and innovative avenues for treating GBM, leveraging therapeutic pathways inaccessible to conventional modalities. These viruses reshape the tumor microenvironment (TME) by “being the matches that light the fire” to bring immunological heat into otherwise cold tumors, as quoted from Webb et al. (2023) [179]. However, even promising approaches like OVT encounter limitations when used as a monotherapy. For instance, intratumoral injection of OVs triggers the recruitment of tumor-associated macrophages (TAMs), which cluster around the injection site, and restrict viral spread into other tumor areas [180]. TAMs may also secrete TNF-α, which suppresses viral replication by inducing apoptosis in infected glioma cells [180]. Furthermore, GBM’s increased stiffness extracellular matrix creates an additional structural barrier that restricts the infiltration of OVs into the tumor [181]. Tumor heterogeneity further complicates treatment efficacy; Sottoriva et al. demonstrated that different regions within the same GBM tumor harbor genetically distinct subclones belonging to separate cellular lineages, making it unlikely that a single therapy could effectively target all tumor populations [182]. Additionally, glioma-initiating cells (GICs) are known for their robust DNA repair mechanisms and self-renewal capacity that exhibit resistance to OVs [183].

The aforementioned limitations of OVT as a standalone therapy along with the limitations of other monotherapies underscore the need for combination therapies, each targeting different facets of the tumor to fully ignite a durable antitumor response. In the following subsections, we will explore key combination strategies involving OVs for GBM. This includes conventional modalities such as surgical resection, chemotherapy, and radiotherapy, as well as widely utilized immunotherapies, including immune checkpoint inhibitors and transgene-armed OVs.

### 3.1. Combination of OVs with Surgical Resection Strategies

Surgical resection remains the cornerstone of standard-of-care treatment for GBM [184]. A greater extent of tumor removal has been consistently associated with improved survival outcomes [184]. However, the infiltrative nature of GBM, characterized by microscopic, finger-like extensions into surrounding brain tissue, poses significant challenges to achieving gross total resection (GTR), even with advanced preoperative and intraoperative imaging modalities [185]. To address the risk of recurrence from residual, migrating tumor cells, one promising strategy involves delivering OVs directly into the post-resection cavity. These viruses will then selectively infect and replicate within residual tumor cells, potentially reducing the likelihood of recurrence as shown in many studies [186]. This approach was clinically demonstrated in a first-in-human phase I trial by Fares et al., in which an engineered oncolytic adenovirus (NSC-CRAd-S-pk7) was administered into the resection margins. The treatment was well-tolerated, induced immune-mediated anti-glioma activity, and showed encouraging survival outcomes in newly diagnosed patients, particularly those with MGMT-unmethylated tumors [106].

### 3.2. Combination of OVs with Chemotherapeutic Agents

Chemotherapy remains a cornerstone for GBM first-line treatment, with temozolomide being the most commonly employed agent [187]. As an alkylating compound, TMZ exerts its cytotoxic effect by introducing O6-methylguanine lesions through methylation of DNA purine bases [188]. However, its clinical efficacy is often compromised by the elevated expression of the DNA repair enzyme O6-methylguanine-DNA methyltransferase (MGMT), which reverses the damage by removing methyl groups from the O6 position of guanine, thereby promoting resistance to TMZ and limiting its potential [187]. To overcome this resistance, combining chemotherapy with OVs has emerged as a promising strategy. Chemotherapeutic agents can sensitize tumor cells to viral oncolysis by inducing DNA damage or impairing cellular defense mechanisms. Conversely, OVs can enhance the effectiveness of chemotherapy by disrupting DNA repair pathways and selectively lysing therapy-resistant tumor cells. This bidirectional enhancement can augment tumor cell death more than would any modality alone [189].

### 3.3. Combination of OVs with Radiotherapy

Radiotherapy (RT) remains a key component of standard GBM treatment. It induces DNA damage, such as strand breaks and base modifications, and contributes to immune activation by promoting the release of proinflammatory cytokines and recruiting effector T cells to the tumor microenvironment [190]. However, the therapeutic efficacy of RT is often limited by the hypoxic nature of GBM. Oxygen plays a critical role in amplifying the effects of low linear energy transfer radiation by stabilizing DNA radicals into permanent damage. In hypoxic conditions, the absence of sufficient oxygen allows tumor cells more time to repair radiation-induced DNA lesions, thereby diminishing treatment effectiveness [191]. Emerging evidence supports the synergistic potential of combining RT with OVs. Studies have shown that ionizing radiation might enhance the replication of oHSV, while the virus, in turn, interferes with DNA double-strand break repair in irradiated GBM cells [59,189]. Furthermore, a recent in vitro study demonstrated that the oncolytic vaccinia virus ΔF4LΔJ2R effectively replicates in and kills irradiated brain tumor–initiating cells [148]. When used in combination with RT in immune-competent orthotopic CT2A-luc mouse models, this approach significantly prolonged survival and achieved complete tumor regression in the majority of treated animals—outperforming each therapy alone [148].

### 3.4. Combination of OVs with Immune Checkpoint Inhibitors

Immune checkpoints such as PD-1, CTLA-4, LAG3, and TIM3 are essential regulators of T cell responses, maintaining immune tolerance and protecting against autoimmunity. However, many tumors, including GBM, hijack these inhibitory pathways to evade immune attack and promote tumor progression [192]. In GBM, immune evasion is facilitated by the widespread expression of PD-L1, which is found in up to 90% of tumor cells. This contributes to T cell exhaustion and suppression of effective immune responses [193]. Such findings have driven the development of immune checkpoint inhibitors (ICIs) which are antibodies designed to block these negative signals and reactivate T cells, enabling them to recognize and eliminate tumor cells [194].

Despite their clinical success in several malignancies, ICIs have shown limited benefit for GBM. The phase III CheckMate 143 trial, for example, reported no significant survival improvement with the PD-1 blocker nivolumab compared to bevacizumab in patients with recurrent GBM [195]. According to Arrieta et al., this resistance is largely attributed to GBM’s low immunogenicity, limited infiltration by effector T cells, and an immunosuppressive TME. Additional obstacles include impaired antigen presentation due to low MHC I and II expression, and the limited permeability of the BBB [196]. To enhance ICI efficacy in GBM, combination strategies with OVs are being explored. OVs can induce immunogenic cell death, promote infiltration of immune cells into the tumor, and increase the expression of immune checkpoint ligands such as PD-L1 that may sensitize tumors to ICIs [137]. This synergistic interaction between OVs and ICIs may help overcome the resistance seen with GBM monotherapy. For instance, in a preclinical study by Sugawara et al. (2021), the combined use of intratumoral G47Δ, a modified HSV, and systemic anti-CTLA-4 antibody significantly depleted regulatory T cells and enhanced the activity of other immune cells highlighting the potential of such combination approaches in reprogramming the GBM immune environment [197].

### 3.5. Arming OVs with Therapeutic Transgenes

The genes for cytokines (GM-CSF, IL-12), immunomodulatory factors (anti-PD1), anti-angiogenic factors, TME inhibitors/degraders, and cytotoxic proteins can now be incorporated into OVs, allowing them to induce potent antitumor immune responses in addition to their traditional role of direct tumor cell lysis [198]. Arming OVs with these transgenes offers two major advantages over unmodified (unarmed) OVs. First, while unarmed OVs primarily rely on direct oncolysis and passive stimulation of the immune system, transgene-armed OVs actively reshape the highly immunosuppressive TME of GBM in multiple targeted mechanisms, leading to more enhanced antitumor immune responses [199]. For example, a modified herpes simplex virus (G47-mIL12) engineered to express IL-12 significantly prolonged survival in mice with intracerebral glioma by targeting glioma stem cells, increasing IFN-γ production, inhibiting angiogenesis, and reducing regulatory T cell infiltration into the tumor [200]. Secondly, arming OVs with cytokines such as IL-12 enables direct delivery of these potent immunostimulatory agents into the tumor, reducing the systemic toxicity typically seen when administered systematically without OVs carriers. This approach boosts the local immune attack on tumor cells while enhancing safety [201,202].

GM-CSF is another key cytokine used to enhance the immunogenic potential of OVs. It recruits and activates dendritic cells, macrophages, and other antigen-presenting cells, ultimately enhancing tumor antigen presentation and priming of T cells [199]. The effectiveness of incorporating GM-CSF into OVs is well demonstrated by talimogene laherparepvec (T-VEC), a genetically engineered HSV-1 designed to express GM-CSF [203]. Clinical trials showed that T-VEC achieved higher response rates than treatment with GM-CSF alone, emphasizing the added therapeutic benefit of coupling OVT with transgene therapy [203].

## 4. Non-Viral Oncolytic Strategies

### 4.1. Oncolytic Bacteria

The GBM is notoriously resistant to treatment due to its hypoxic, necrotic microenvironment, which limits the effectiveness of chemotherapy and radiotherapy [204,205]. However, this same environment makes it an ideal target for anaerobic bacteria like *Clostridium novyi*-NT, a genetically modified strain engineered to lack its lethal toxin gene [206]. Non-viral oncolytic strategies have garnered attention as alternative therapeutic avenues for GBM, particularly due to their ability to exploit the tumor’s hypoxic microenvironment and stimulate anti-tumor immunity. Among these, *Clostridium novyi*-NT (*C. novyi*-NT), which is an attenuated strain of an obligate anaerobic bacterium, has shown remarkable tumor specificity and therapeutic efficacy in preclinical GBM models.

When injected directly into tumors, *C. novyi*-NT spores germinate only in hypoxic regions, selectively destroying tumor tissue while sparing healthy brain (Figure 3) [161]. In rat glioma models, intratumoral injection led to rapid tumor reduction and significantly improved survival. Importantly, the bacteria could also target invasive tumor cells beyond the resection margin, which are areas typically inaccessible to conventional therapies. Beyond direct tumor lysis, bacterial germination triggers strong local inflammation and may stimulate systemic antitumor immune responses [206,207]. Compared to systemic delivery, direct injection ensures better tumor targeting with fewer off-target effects [208].

These findings were mirrored in a veterinary trial: dogs with soft tissue sarcomas responded well to *C. novyi*-NT, with a 37.5% objective response rate and manageable local side effects [209]. These encouraging findings justified compassionate use in a human patient with leiomyosarcoma, where a single intratumoral injection significantly reduced tumor volume, supporting the feasibility of bacterial-mediated tumor lysis in solid malignancies [206]. Given the infiltrative growth, hypoxia, and resistance to standard therapies seen in high-grade gliomas, localized bacterial therapy such as *C. novyi*-NT may offer a targeted and effective strategy, albeit with potential risks like brain edema or abscess formation that require clinical monitoring and surgical readiness.

Another promising approach involves VXM01, an oral DNA vaccine based on Salmonella typhi Ty21a engineered to carry a plasmid encoding vascular endothelial growth factor receptor 2 (VEGFR-2) (Figure 3). This strategy aims to elicit systemic T-cell responses targeting tumor vasculature. In a clinical trial involving patients with progressive GBM who had failed standard radiochemotherapy, VXM01 was well tolerated and induced VEGFR-2–specific T-cell responses in over half of treated individuals [210]. Notably, enhanced CD8+/Treg ratios and reduced intratumoral PD-L1 expression were observed post-treatment, correlating with prolonged survival in some patients. These findings support further development of VXM01, especially in combination with immune checkpoint inhibitors such as anti-PD-L1 antibodies, to enhance immune activation in an otherwise immunosuppressive GBM microenvironment.

Recent research has highlighted the promising role of bacteria as vectors for therapeutic gene delivery or to secrete anti-tumor agents locally, minimizing systemic toxicity [211]. In addition, bacterial components such as lipopolysaccharides or flagellin can stimulate innate immune responses and reverse the immunosuppressive tumor microenvironment, enhancing the efficacy of immunotherapies [212]. Some studies have even explored the use of bacteria to alter the gut–brain axis, indirectly influencing the brain’s immune surveillance and tumor response. Taken together, these findings suggest that with further refinement and clinical translation, bacteria-based strategies could offer a novel and complementary approach to conventional GBM therapies.

Gram-negative bacteria, such as attenuated Salmonella typhimurium (VNP20009), can stimulate innate immunity through Toll-like and NOD-like receptors on phagocytes, reactivating antitumor responses in the post-surgical tumor bed [213]. These bacteria promote immune cell infiltration, overcome immune “coldness,” and present tumor-associated antigens to T cells, bridging innate and adaptive immunity [214,215]. Intriguingly, retrospective studies suggest improved GBM patient outcomes when localized Gram-negative infections occur postoperatively, indicating a therapeutic window for bacterial intervention [216,217,218]. Building on this, engineered bacteria, such as immunostimulatory autolysing Salmonella-nanocapsule delivery systems (IASNDS), can safely colonize hypoxic tumor regions and release immunogenic bacterial components through controlled lysis, inducing pyroptosis and antigen release [211]. These events enhance antigen presentation, cytokine release, and T cell activation, culminating in robust antitumor immunity. Coupled with ATP-responsive hydrogels that sustain local immune activation via CpG oligonucleotide release, this innovative bacteriotherapy strategy demonstrates significant promise in preventing GBM recurrence by reshaping the immune microenvironment, mobilizing both innate and adaptive immunity, and offering a transformative post-surgical adjunct in GBM management.

Looking forward, the integration of CRISPR-Cas9 genome editing offers a transformative avenue to enhance the precision, safety, and efficacy of oncolytic bacterial therapies. CRISPR can be employed to knock out virulence genes, insert safety switches, or engineer bacterial payloads that secrete immune-stimulating cytokines (e.g., IL-12, GM-CSF), tumor-selective toxins, or checkpoint inhibitors directly within the tumor microenvironment [219]. For example, Clostridium or Salmonella strains could be modified to include inducible suicide genes or biosensors responsive to tumor-specific signals, allowing for precise control over bacterial replication and therapeutic payloads. Additionally, CRISPR-based engineering enables the creation of multi-functional bacterial vectors that combine tumor targeting, immune modulation, and biosafety; all essential for clinical translation for GBM, where immune privilege and anatomical constraints pose significant therapeutic barriers.

### 4.2. Oncolytic Peptides

In recent years, peptide-based therapies have emerged as a promising alternative for GBM as a non-viral oncolytic strategy. Peptides demonstrate high specificity in targeted therapies due to their ability to bind with high affinity to receptors that are overexpressed or tumor-specific in cancer cells. Thanks to these properties, they can selectively target tumor cells while exerting minimal toxic effects on healthy tissues. Additionally, certain peptides have been reported in the literature to exhibit intrinsic antineoplastic activity and may be used as pharmacological agents in drug delivery systems or for modulating intracellular signaling cascades. Among these strategies, p28, a peptide with high permeability across cellular and blood–brain barriers, stands out for its therapeutic efficacy against GBM as well as its potential role as a carrier for diagnostic imaging agents. In GBM animal models, p28 has been shown to enhance the cytotoxic activity of temozolomide, a DNA-damaging agent known to be effective against brain tumors. Furthermore, the chemical conjugate of p28 with the near-infrared (NIR) dye indocyanine green (ICG), referred to as ICG-p28, has been demonstrated to selectively localize to tumors in GBM animal models, with accumulation observed in both the cytoplasmic space and the nucleus. Most notably, p28 has exhibited a significant anti-tumor effect on GBM [220]. Collectively, these findings suggest that p28 possesses strong potential as a dual-function agent, capable of both therapeutic impact and precision imaging for GBM management.

The tumor microenvironment plays a critical role in mediating resistance to GBM therapies. As a result, recent therapeutic strategies have focused on targeting components of the extracellular matrix (ECM), which contributes to this resistance. Brevican (Bcan), an ECM protein highly expressed in GBM tissue, has a specific isoform called dg-Bcan that was recently identified as a promising molecular target for GBM. In this context, a newly designed 8-amino-acid D-peptide, termed BTP-7, demonstrates strong binding affinity and high specificity toward dg-Bcan. It is selectively taken up by GBM cells, allowing for precise tumor targeting. To demonstrate its diagnostic potential, BTP-7 was radiolabeled with ^18^F and used in PET imaging in mouse models bearing intracranial GBM. The imaging results confirmed that BTP-7 effectively localizes to tumor regions, supporting its use in non-invasive tumor detection. Furthermore, BTP-7’s structure, which is resistant to enzymatic degradation, allows it to cross the blood–brain barrier and accumulate within tumor tissue [221]. These properties suggest that BTP-7 may serve as a promising tool for both diagnostic imaging and targeted therapy in GBM, offering a minimally invasive approach to managing this highly aggressive cancer.

In addition to preclinical studies investigating the effects of oncolytic peptides in GBM treatment, clinical trials have also assessed the efficacy of peptide-based vaccines in GBM patients. A recent study followed 173 patients with GBM who were treated with a personalized peptide vaccine targeting tumor-specific neoantigens and evaluated the feasibility, immunogenicity, and therapeutic potential of this individualized vaccine approach [222]. Among newly diagnosed and recurrent patients, the median overall survival from the time of diagnosis was reported as 31.9 months. The peptide vaccine was well tolerated, even when administered alongside standard therapies. Serious adverse events were rare, and side effects were mostly mild to moderate in severity. Blood sample analysis revealed that the vaccine elicited strong T-cell responses in a large proportion of patients. Furthermore, these vaccine-induced T-cell responses were found to be durable in most individuals. Patients who developed strong T-cell responses against the peptides demonstrated significantly longer survival [222]. In conclusion, personalized neoantigen-based peptide vaccines emerge as an effective and well-tolerated immunotherapeutic strategy for patients with glioblastoma.

## 5. Concerns and Future Directions in Oncolytic Therapies

Several challenges have been proposed to explain the limited success of OVs in the treatment of GBMs. These include suboptimal viral delivery methods, issues related to prodrug selection and compliance, and an insufficient induction of durable antitumor immune responses [179]. Most studies to date have employed intratumoral injection of viral particles following surgical resection. Although this strategy enables direct delivery to the tumor site, it remains difficult to ascertain the extent to which residual microscopic tumor cells are exposed to the virus, particularly in the presence of normal brain parenchyma or necrotic tissue, which may act as physical barriers [223]. Furthermore, the efficiency of viral uptake by tumor cells and the likelihood of successful replication remain uncertain. Enhancing viral tropism for tumor cells and optimizing delivery strategies such as through convection-enhanced delivery may help address these limitations [224]. Although techniques such as convection-enhanced delivery may improve distribution, inconsistent prodrug dosing and timing, as observed in the Toca 511/Toca FC trial, where patients received fewer cycles than preclinical models suggested were necessary, further limit therapeutic efficacy [223].

The profound intertumoral and intratumoral heterogeneity of GBM (well-characterized by TCGA into proneural, mesenchymal, and classical subtypes) complicates treatment, as phenotypic plasticity and subtype transitions can lead to resistance and failure to target all tumor cell populations [225,226,227]. Moreover, the TME, composed of the extracellular matrix, immune cells, and stem cells, significantly influences tumor progression and therapeutic response, and must be considered both as a barrier and a potential modulator of efficacy [196]. Perhaps the greatest obstacle is the profoundly immunosuppressive nature of GBM, characterized by a low mutational burden, secretion of immunosuppressive factors [228], sparse and exhausted tumor-infiltrating lymphocytes [229], and systemic T cell sequestration [228]. Therefore, achieving lasting efficacy with OVs will likely require combination strategies, such as pairing OVs with immune checkpoint inhibitors or cytokine-based therapies to boost peritumoral CD8+ lymphocyte infiltration [228,229].

Future directions in oncolytic therapies of GBM are increasingly focused on improving the specificity, delivery, and overall therapeutic efficacy of these agents. One promising approach involves nanoparticle-based delivery systems, which can encapsulate oncolytic viruses or agents to help them bypass biological barriers such as the blood–brain barrier. Another area of development includes the use of armed oncolytic viruses that are genetically modified to deliver therapeutic payloads, such as cytokines like IL-12 or IFN-beta, or other immune-modulating factors. Additionally, combining oncolytic viruses with targeted therapies such as agents against EGFR, VEGF, or IDH mutations may enhance antitumor effects through complementary mechanisms. Efforts are also underway to identify reliable biomarkers to guide patient selection, aiming to match specific molecular profiles with the most effective therapeutic strategies.

## 6. Conclusions

Oncolytic therapies represent a promising frontier in the fight against GBM, offering a novel approach that combines targeted tumor lysis with potent immune activation. Despite significant strides such as the clinical progress of HSV-based therapies, challenges remain, including immune evasion, tumor heterogeneity, and delivery limitations imposed by the blood–brain barrier. Integrating oncolytic viruses with immune checkpoint inhibitors, chemotherapy, and radiation has demonstrated synergistic potential, but these strategies require further clinical validation. Moreover, non-viral oncolytic agents, such as tumor-targeting bacteria and synthetic peptides, remain underexplored, but they may expand the therapeutic arsenal. Future research must prioritize refining delivery techniques, enhancing tumor selectivity, and leveraging biomarkers for patient stratification. By addressing these gaps and aligning OVT with precision medicine strategies, there is a real opportunity to overcome the therapeutic stagnation in GBM and offer patients more effective, durable treatment options.

## Figures and Tables

**Figure 1 cancers-17-02550-f001:**
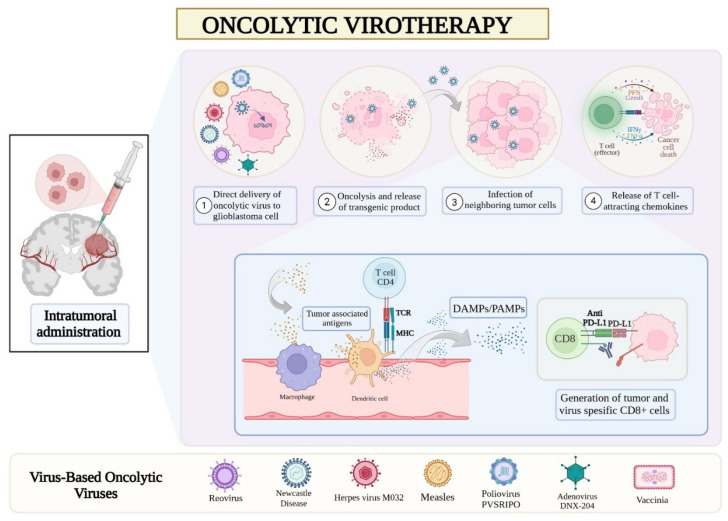
This schematic illustrates the immunologic and cytolytic mechanisms of oncolytic virotherapy against glioblastoma. Following intratumoral administration, oncolytic viruses selectively infect tumor cells (1), replicate and induce oncolysis with release of transgenic and viral products (2), and spread to neighboring tumor cells (3). Viral infection triggers the release of chemokines that attract immune cells (4), leading to antigen presentation by macrophages and dendritic cells. This promotes activation of tumor- and virus-specific CD4^+^ and CD8^+^ T cells, while anti–PD-L1 checkpoint blockade enhances cytotoxic T-cell function. Several viral platforms, including poliovirus (PVS-RIPO), adenovirus (DNX-2401), herpes simplex virus (M032), and measles virus, are under clinical investigation for glioblastoma therapy [created with BioRender.com].

**Figure 2 cancers-17-02550-f002:**
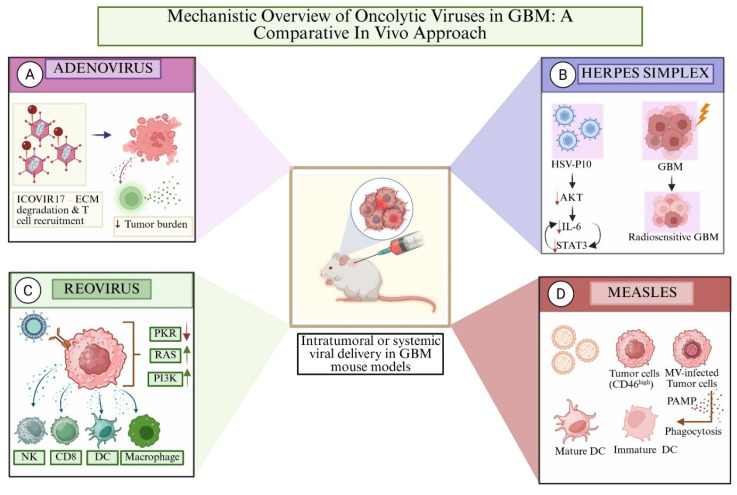
This figure illustrates the distinct mechanisms by which various oncolytic viruses modulate the glioblastoma (GBM) microenvironment in vivo. (**A**) Adenovirus (ICOVIR17) promotes extracellular matrix degradation and T-cell recruitment, reducing tumor burden. (**B**) Herpes simplex virus (HSV-P10) downregulates the AKT/IL-6/STAT3 pathway, sensitizing GBM cells to radiotherapy. (**C**) Reovirus selectively replicates in RAS-activated tumor cells by suppressing PKR, activating NK cells, CD8^+^ T cells, macrophages, and dendritic cells. (**D**) Measles virus infects CD46-high tumor cells, releasing PAMPs and enhancing phagocytosis by dendritic cells, particularly promoting maturation. Viruses are delivered intratumorally or systemically in GBM-bearing mice in all models, offering insight into their therapeutic and immunomodulatory potential. [Created with BioRender.com].

**Figure 3 cancers-17-02550-f003:**
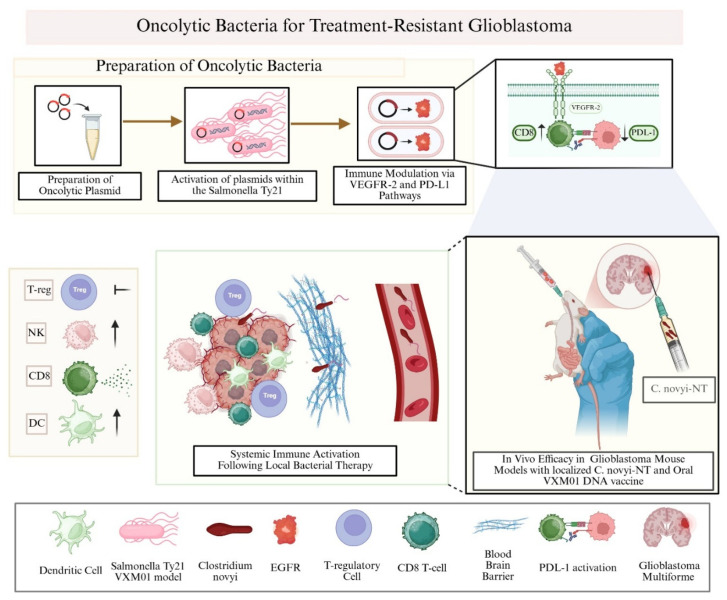
Schematic representation of engineered oncolytic bacteria targeting glioblastoma. VXM01, an oral Salmonella-based VEGFR-2 DNA vaccine, enhances CD8^+^ T-cell responses and reduces PD-L1 expression. Locally injected *C. novyi*-NT germinates in hypoxic tumor cores, inducing direct oncolysis and immune activation. Both approaches remodel the tumor microenvironment and elicit systemic antitumor immunity. Preclinical models and early clinical data support their translational potential in GBM therapy. [Created with BioRender.com].

**Table 1 cancers-17-02550-t001:** Overview of Oncolytic Herpes Simplex Virus (oHSV) Variants and Therapeutic Strategies in Glioblastoma Models.

oHSV Variant	Genetic Modifications/Additions	Study Model	Key Findings	Reference
z	Lacks thymidine kinase (TK); attenuated neurovirulence	U87 glioma cells (in vitro, mice)	Cytotoxic to human glioma cells; improved survival and reduced tumor burden in mice	[29]
oHSV-P10	Expresses PTEN-L	Human GBM12 cells and mice	Eliminated GSCs, blocked IL6/JAK/STAT3 signaling; overcame radiotherapy resistance	[30]
HSV-P10	Expresses PTEN-α	GBM cells, immune-competent mice	Suppressed PI3K/AKT pathway, reduced PD-L1, enhanced adaptive immune response	[31]
oHSV-1 (ICP4+)	Deleted γ34.5 and ICP47; retains ICP4	Glioma cell lines and animal models	Reduced invasion via downregulation of Sp1; high anti-tumor activity with less immune activation	[32]
EGFP-oHSV-1	Δγ34.5/ΔUS12; includes EGFP reporter	GL261 glioma mouse model	Improved survival and immune response; greater tumor control vs. other oHSV-1 variants	[33]
C5252	Deleted γ34.5 and 15 kb repeat; expresses IL-12 and anti-PD-1 fragment	Human GBM cells and mice	Strong cytotoxicity despite low replication; induced apoptosis, immune activation (IFN-γ, TNF-α)	[34]
G47Δ-mIL2	Expresses IL-2	Orthotopic glioma mouse model	Enhanced CD4+/CD8+ T-cell infiltration; prolonged survival; no synergy with PD-1 blockade	[35]
G47Δ-mIL12	Expresses IL-12; combined with anti-PD-1/CTLA-4	Glioma mouse model	Improved survival; increased M1 macrophages and effector T cells; enhanced immune response	[36]
R-115	Targets HER2; expresses IL-12	HER2+ glioma in BALB/c mice	Long-term remission, antibody generation, HER2-independent protection; no CD4/CD8 increase noted	[37]
R-613	Targets EGFRvIII	hGICs and EGFRvIII+ glioma in mice	Effective in early-stage tumors; less effective in advanced disease; suitable for combination therapy	[38]
OH2 (HSV-2 based)	HSV-2-derived oncolytic virus with tumor-selective replication	In vitro, xenograft mice	Selective tumor replication, DNA damage, suppressed tumor cells and M2 TAMs, enhanced macrophage and T cell infiltration, slowed GBM, prolonged survival	[39]
NG34 (HSV-1 with GADD34)	HSV-1 engineered to express GADD34 under nestin promoter	In vitro, mouse models	Similar efficacy to rQNestin34.5, reduced neurotoxicity and brain damage	[40]
HSV + CAR-T cells (B7-H3-directed)	CAR-T cells engineered to express B7-H3-targeted CAR and carry HSV	Orthotopic GBM mice	CAR-T cells deliver HSV to distant tumor sites, enhanced T cell infiltration, prolonged survival	[41]
CAR-T + HSV-1 G47Δ	HSV-1 G47Δ combined with CAR-T cells	Mouse models	Tumor regression, extended survival	[42]
oHSV-1	Wild-type or engineered oHSV-1	Murine models, in vitro	TNFα from M1 macrophages/microglia inhibits viral replication; blocking TNFα increases viral spread and survival	[43]
MEK inhibitor + oHSV-1	Pharmacological MEK inhibitor combined with oHSV-1	In vitro, murine glioma	Trametinib reduces TNFα production, enhances oHSV replication and survival	[44]
oHSV + Gamma Secretase Inhibitor (GSI)	Wild-type oHSV combined with NOTCH pathway inhibitor	In vitro, orthotopic mice	Inhibition of NOTCH signaling boosts therapy efficacy with maintained safety	[45]
oHSV G47Δ	HSV-1 G47Δ	Patient-derived cultures, mice	Fasting boosts viral replication and cytotoxicity by suppressing JNK pathway	[46]
γ34.5-deficient oHSV	Deletion of γ34.5 gene in oHSV; Us11 protein expression	GSCs, ScGCs culture	GSCs resist virus replication due to translational blockade; Us11 protein restores replication	[47]
Bortezomib + oHSV1 + NK cells	oHSV + proteasome inhibitor + NK cells	In vitro, mice	Necroptosis induction, increased NK activation, improved tumor suppression	[48]
CCN1 and HSV resistance	Wild-type HSV-1; CCN1 expression modulates resistance	GBM cell lines	CCN1 mediates early viral resistance by innate immune activation	[49]
oHSV + TGF-β receptor inhibitor	oHSV combined with TGF-β pathway inhibitor	GSCs, animal models	Blocking TGF-β boosts viral replication and survival via JNK-MAPK pathway	[50]
oHSV + C16 inhibitor	Pharmacologic STAT1/3 inhibitor combined with oHSV	U87 xenograft, cell cultures	Inhibiting STAT1/3 in microglia/macrophages promotes oHSV replication and tumor regression	[51]
oHSV releasing ChaseM enzyme + temozolomide	oHSV engineered to release ChaseM enzyme (degrades CSPGs)	Preclinical mouse models	Degrades CSPGs, improves tumor penetration and apoptotic death, extends survival	[52]
oHSV + anti-HMGB1 antibodies	Wild-type oHSV combined with HMGB1 neutralizing antibodies	In vitro, mouse brain tumors	Blocking HMGB1 reduces edema and improves survival with oHSV therapy	[53]
KG4:T124	HSV-1 derivative (specific modifications not detailed)	GL261N4 and CT2A murine glioma models	Cleared quickly in CT2A; limited immune response and therapeutic effect.	[54]
rQNestin34.5v.1	ICP34.5 under nestin promoter	GL261N4 and CT2A murine glioma models	Higher viral load; persisted longer in GL261N4, enhancing immune infiltration and survival.	[55]
CXCR4-targeted oHSV	Glycoprotein D modified with CXCR4-specific nanobody in attenuated HSV-1	GSC xenograft mouse models	Targeted CXCR4+ GSCs; reduced tumor growth and improved survival.	[56]

oHSV: Oncolytic Herpes Simplex Virus, TK: Thymidine Kinase, GBM: Glioblastoma, GSCs: Glioma Stem Cells, IL: Interleukin, JAK: Janus Kinase, STAT: Signal Transducer and Activator of Transcription, PI3K: Phosphoinositide 3-Kinase, AKT: Protein Kinase B, PD-L1: Programmed Death-Ligand 1, Sp1: Specificity Protein 1, EGFP: Enhanced Green Fluorescent Protein, IFN-γ: Interferon gamma, TNF-α: Tumor Necrosis Factor alpha, CTLA-4: Cytotoxic T-Lymphocyte Antigen 4, HER2: Human Epidermal Growth Factor Receptor 2, EGFRvIII: Epidermal Growth Factor Receptor variant III, TAMs: Tumor-Associated Macrophages, CAR-T: Chimeric Antigen Receptor T-cells, JNK: c-Jun N-terminal Kinase, STAT1/3: STAT1 and STAT3 Proteins, CSPGs: Chondroitin Sulfate Proteoglycans, HMGB1: High Mobility Group Box 1.

**Table 2 cancers-17-02550-t002:** Summary of Clinical Studies Evaluating oHSV-Based Treatments in Glioblastoma Patients.

Virus Used	Patient Population	Design and Intervention	Adverse Events	Median Survival	Notable Findings	Reference
G207 (γ134.5-deleted HSV-1)	9 adults with recurrent malignant glioma	Single intratumoral G207 injection + 24 h later 5 Gy radiation; 2 patients had a second injection	Well tolerated; no severe side effects	7.5 months	Safe combination with radiotherapy; 3 patients showed marked radiologic response	[57]
G207	12 pediatric patients (7–18 y/o), mostly GBM	Intratumoral G207 ± radiation	High rate of AEs (e.g., diarrhea, bradycardia, seizures), but manageable	12.2 months	11/12 had clinical or radiological improvement; increased lymphocyte infiltration; 4 survived > 18 months	[58]
G47∆ (triple-mutated oHSV)	13 adults with recurrent/advanced GBM	Up to 6 intratumoral injections over 2 weeks	Common: nausea, fever, headache; manageable with corticosteroids	Not reported; 3 > 46 mo	CD4+/CD8+ T-cell infiltration; 1 patient survived > 11 years	[59]
M032 (IL-12 expressing HSV-1)	21 adults with recurrent glioma	Single intratumoral injection	Grade 3–4 AEs in only 1 patient at high dose; no severe toxicity at max dose	9.38 months	Generally well tolerated; individualized response suggests need for personalized dosing	[60]
CAN-3110 (ICP34.5 under nestin promoter)	41 patients with recurrent GBM	Single intratumoral injection	No serious AEs at highest doses	Not stated clearly	HSV-seropositive patients had better survival; T-cell activation and immune gene upregulation observed	[61]

oHSV: Oncolytic Herpes Simplex Virus, GBM: Glioblastoma, IL: Interleukin, AE(s): Adverse Event(s), CTLA-4: Cytotoxic T-Lymphocyte Antigen 4, CD4+/CD8+: T-cell Surface Markers (Helper/Cytotoxic), MRI: Magnetic Resonance Imaging, Gy: Gray (Unit of Radiation Dose).

**Table 3 cancers-17-02550-t003:** Overview of Oncolytic Adenovirus Variants and Therapeutic Strategies in Glioblastoma Models.

oADV Variant	Genetic Modifications/Additions	Study Model	Key Findings	Ref.
Replicating vs. Non-replicating oAdVs	Replication-competent vs. incompetent adenoviruses	Cell lines, mouse models	Replicating oAdVs enhanced immune cell infiltration and survival	[69]
Ad5-pIX-Ad37	IX capsid protein with dimerization domain; fiber knob from Ad37	In vitro, in vivo	Enhanced cell entry and oncolytic activity	[70]
oAdV-ApoA1	Carries apolipoprotein A1	Cell lines, mouse models	Reduced 7-KC, activated TNF signaling, improved immune response	[71]
ICOVIR17	Carries hyaluronidase enzyme	GBM mouse models	Increased macrophages, CD8+ T cells, and survival with anti-PD-1	[72]
ICOVIR15	∆24-E1A, RGD-modified fiber	GBM cells, mouse models	Targeted FAP+ pericytes and tumor cells; induced apoptosis	[73]
ONCOTECH	T cell–associated, PD-L1 targeting	Cancer mouse models	Reduced PD-L1, enhanced survival	[74]
Ad5-Ki67/IL-15	Ki67 promoter; IL-15 expression	Glioma cells, mouse models	Reduced PD-L1, boosted T cell infiltration	[75]
MSC-Ad5-Ki67/IL-15	MSC-carried virus with IL-15 and Ki67 promoter	In vitro, in vivo	Enhanced macrophage infiltration and efficacy	[76]
TS-2021 (Ad5 KT-E1A-IL-15)	Ki67 promoter, TGF-β2 5′UTR, IL-15	In vitro, in vivo	Reduced tumor burden, improved survival	[77]
TS-2021 + Olaparib	Same as TS-2021; combined with PARP inhibitor	GBM cells, mouse models	Synergistic tumor apoptosis and survival benefit	[78]
XVir-N-31 (Intranasal)	Carrier-cell optimized oAdV	GBM-bearing mice	Non-invasive delivery reduced tumor burden, improved survival	[79]
XVir-N-31 + ICI	Combined with anti-PD-1/PD-L1	In vitro, humanized mouse models	Enhanced immune cell infiltration, tumor regression	[80]
YSCH-01	Recombinant interferon-like gene	Glioma cells, hamster models	Strong local and distant tumor suppression	[81]
H5CmTERT-Ad/TRAIL	hTERT promoter, secretable trimeric TRAIL	In vitro, in vivo	Effective in TRAIL-resistant tumors, induced death in hypoxia	[82]
Ad6	Native Ad6 serotype	GBM cells, mouse models	Cytotoxicity, reduced GBM stem cells	[83]
Ad5 (hTERT/survivin promoters)	GBM-specific promoters (hTERT, survivin)	GBM cell lines	Selective cytotoxicity in GBM cells	[84]
Delta-24-RGD (Proteomic analysis)	∆24-E1A, RGD-modified fiber	Phase I clinical trial samples	Altered kinase/cytokine profiles, immune activation	[85]
CTV (Ad3 fiber + Ad5 capsid)	Produces MDA-7/IL-24	In vivo, GBM models	Extended survival, enhanced with Temozolomide	[86]
PD-BM-MSC-D24	Delta-24-RGD loaded in chemo-treated BM-hMSC	In vitro, in vivo	Effective delivery and tumor suppression	[87]
Delta-24-RGDOX	Oncolytic adenovirus with RGD motif and OX40L	Mouse models	Prolonged survival; changes in gut microbiota with dominance of *Bifidobacterium*. Microbiota may influence therapeutic efficacy.	[88]
Delta-24-GREAT	Delta-24 modified with GITRL gene	Human and mouse glioma cell lines; mouse models	Enhanced immune response, increased memory T cells, tumor rejection after re-challenge.	[89]
Delta-24-RGD + HDAC inhibitors	Combined with scriptaid and LBH589	Patient-derived glioblastoma lines	Synergistic antitumor effects; scriptaid ↑ caspase-3/7 and apoptosis; LBH589 ↑ LDH and phospho-p70S6K.	[90]
AdCMVdelta24	Delta-24 under CMV promoter	Mouse GBM models	Reduced Tregs, increased IFNγ+ CD8+ T cells; reprogrammed Tregs into stimulatory phenotype.	[91]
hMSC-D24	Delta-24-RGD loaded into human MSCs	Dog GBM model (large animal); intra-arterial delivery (ESIA)	ESIA safe in anterior cerebral circulation; stroke risk in posterior; proof-of-concept for large-animal delivery method.	[92]
Ad5/35-delta-24, Ad5/3-delta-24	Fiber region modified	Human and rodent glioma lines; mouse models	Ad5/35-delta-24: strong immune-mediated tumor suppression; induced immune memory; superior to Ad5-delta-24-RGD.	[93]
CAN-2409 + dexamethasone	Simultaneous administration with corticosteroid	In vitro and in vivo experiments	decreased immune activation; decreased tumor response; decreased median survival; dexamethasone suppresses CAN-2409 efficacy.	[94]
Delta-24-RGD + anti-PD-1	Combination with immune checkpoint blockade	In vivo and in vitro models	Synergistic effect; Increased CD8+ T cells and IFNγ production; improved survival over monotherapies.	[95]
H101 + anti-PD-1	H101 suppresses CD47; anti-PD-1 immunotherapy	Human glioblastoma lines (U87-MG)	Increased T cell infiltration, macrophage phagocytosis, cytokines; enhanced anti-tumor effect.	[96]
CAN-2409 + ATR inhibitor (AZD6738)	Combination with DNA damage repair inhibitor	In vitro and in vivo GBM models	Increased γH2AX; decreased PD-L1; improved survival; enhanced DNA damage and immune response.	[97]
Oncolytic Ad (receptor sensitivity study)	N/A	Human glioma cell lines (Grade II–IV)	Receptor expression (CAR, CD46, DSG-2) not predictive of infectivity or efficacy.	[98]
Delta-24-RGD + TMZ	Combination with temozolomide (standard chemo)	Murine glioma lines; mouse models	Synergy when Delta-24-RGD precedes TMZ; reversed effects if sequence is changed; Increased CD8+ T cells.	[99]
Ad5-Delta-24-RGD with L3-23K vs. L5-Fiber gene addition	Gene insertion at different viral regions	GBM cell lines; mouse models	Gene expression higher at L3-23K; insertion site affected oncolytic activity; no major difference in cytotoxicity.	[100]
CXCL11-carrying oAd	oAd encoding CXCL11 chemokine	Orthotopic GBM mouse models; cell cultures	Increased CD8+ T cell activation; Increased Tregs; improved CAR-T therapy efficacy in GBM.	[101]
oAd-IL7 + CAR-T (B7H3)	Oncolytic adenovirus expressing IL-7	In vivo and in vitro models	Increased intratumoral T cells; Increased median survival; synergistic effect with CAR-T therapy.	[102]
OA@TA-Fe^3+^-CXCL11 oAd	CXCL11 oAd coated with tannic acid & Fe^3+^ ions	In vivo and in vitro GBM models	Increased retention and oncolytic activity; Fe^3+^ reduced hypoxia via O_2_ generation; immune stimulation.	[103]
NSC.CRAd-S-pk7	CRAd-Survivin-pk7 delivered via neural stem cells	Mice with competent immune system	Multiple doses at high levels effective; immune system did not hinder therapy; shown to be safe in Phase I trials.	[104]
CAN-2409	Expresses HSV-TK (thymidine kinase)	Glioma stem-like cells; mouse models	Enriched p53/cell cycle pathways; regulated MYC, CCNB1, PLK1, CDC20; Increased IL-12, Increased T cell activation.	[105]

oAdV: Oncolytic Adenovirus, GBM: Glioblastoma, IL: Interleukin, PD-L1: Programmed Death-Ligand 1, MSC: Mesenchymal Stem Cell, TGF-β: Transforming Growth Factor beta, TRAIL: TNF-Related Apoptosis-Inducing Ligand, TMZ: Temozolomide, CAR-T: Chimeric Antigen Receptor T-cells, IFNγ: Interferon gamma, CD8+: Cytotoxic T-cell Marker, FAP: Fibroblast Activation Protein, RGD: Arginine-Glycine-Aspartic Acid Motif, CMV: Cytomegalovirus, MYC: MYC Proto-Oncogene, PLK1: Polo-Like Kinase 1, CDC20: Cell Division Cycle Protein 20.

**Table 4 cancers-17-02550-t004:** Summary of Clinical Studies Evaluating Adenovirus-Based Treatments in Glioblastoma Patients.

Virus Used	Patient Population	Design and Intervention	Adverse Events	Median Survival	Notable Findings	Ref.
NSC-CRAd-S-pk7	11 newly diagnosed glioblastoma patients	Phase 1; oAdv delivered via neural stem cells (NSC) after tumor resection; followed by standard chemo-radiotherapy	Decreased lymphocytes, headache, anemia, fatigue, nausea, hypoalbuminemia; NSC-CRAd-S-pk7-related meningitis (1 pt), subdural fluid (1 pt)	18.4 months	Safe; did not delay standard therapy; promising survival outcomes; supports Phase 2 progression	[106]
Delta24-RGD (DNX-2401)	Patients with recurrent glioblastoma	Phase 1; oAdv administered via convection-enhanced delivery (CED) to tumor and peritumoral areas	Dose-limiting: increased intracranial pressure, temporary viral meningitis	129 days	1 complete responder (8-year survival), 1 partial response (2.5 years); ↑ NK, T cells, proinflammatory cytokines; immune activation observed	[107]
Ad5-Δ24.RGD	GBM patients with pre-existing adenovirus antibodies	Observational immune monitoring study	Not specified	Not specified	Despite neutralizing antibodies, RGD modification enabled efficacy by alternative cellular entry; supports oAdv utility in seropositive patients	[108]
DNX-2401 + pembrolizumab	3 recurrent glioblastoma patients	Phase 2 pilot; DNX-2401 delivered via SmartFlow catheter under real-time MRI guidance; followed by pembrolizumab	None reported	Not reported	2 partial responders with infusion area >1 cm; 1 non-responder with <1 cm infusion; safe and feasible approach for image-guided intratumoral administration	[109]
DNX-2401 + pembrolizumab	49 patients with recurrent glioblastoma	Phase 1/2; DNX-2401 combined with anti-PD-1 (pembrolizumab)	No dose-limiting toxicities reported	12.5 months	Combination was safe; MRI response associated with long-term survival; supports benefit over monotherapy	[110]

GBM: Glioblastoma, oAdV: Oncolytic Adenovirus, NSC: Neural Stem Cell, CED: Convection-Enhanced Delivery, AE(s): Adverse Event(s), MRI: Magnetic Resonance Imaging, PD-1: Programmed Death-1, IL: Interleukin, CR: Complete Response, PR: Partial Response.

**Table 5 cancers-17-02550-t005:** Therapeutic Development of oMeV for Glioblastoma: Genetic Engineering, Preclinical Efficacy, and Clinical Safety.

oMeV Variant	Genetic Modifications/Additions	Study Model	Key Findings	Ref.
Preclinical				
MV-CEA	Insertion of the soluble N-terminal extracellular domain of human CEA into MV-NSe backbone (MV-Edm lineage); CEA gene placed before N gene	Human glioma cell lines (primary and GBM-derived); orthotopic glioma mouse models (e.g., U87, GBM14, GBM39, GBM6)	Enabled noninvasive monitoring of viral gene expression via serum CEA levels; induced selective cytopathic effects in glioma cells; promoted apoptosis; spared normal astrocytes and fibroblasts; prolonged survival in glioma models	[113]
MV-miR-122	Engineered to encode microRNA-122 (miR-122); tested with luciferase reporter system	In vitro glioma cell model	Demonstrated that MV may have the potential to deliver functional miRNA and leads to a reduced target protein expression by 40%; limited by Drosha-mediated miRNA processing inefficiency in cytoplasm	[114]
MeV + Ruxolitinib	Combination of wild-type Edmonston-lineage MeV with JAK1 pathway inhibitor ruxolitinib	Patient-derived xenograft (PDX) glioblastoma models; 22-gene expression analysis	JAK1 inhibition increased MeV replication and therapeutic sensitivity; highlighted potential for combination virotherapy and immune modulation	[115]
MeV-stealth	MeV glycoproteins replaced with CDV glycoproteins; CDV-H fused with CD46-targeting scFv; CDV-F signal peptide modified to redirect tropism	Multiple tumor xenografts in immunodeficient mice	Immune-evasive tumor targeting; specific lysis of CD46-overexpressing tumors; reduced off-target effects	[116]
MV-L16 (live attenuated measles virus strain)	GBM-derived primary cell lines (Gbl7n, Gbl11n, etc.)	Preclinical translational study (ex vivo analysis of patient-derived glioma lines)	MV-L16 induced significant caspase-3/7 activation, confirming apoptosis. mRNA-seq showed predictive immunologic gene expression changes in virus-sensitive GBM cells.	[117]
**Clinical Studies**				
**Virus Used**	**Design and Intervention**	**Adverse Events**	**Notable Findings**	**Ref.**
MV-CEA (Edmonston strain expressing carcinoembryonic antigen)	22 patients with recurrent glioblastoma (GBM) Phase I, First-in-human trial (NCT00390299). Group A: MV-CEA injected into resection cavity. Group B: Intratumoral injection Day 1, followed by resection cavity injection Day 5.	No dose-limiting toxicity, even at max dose. Minor, manageable adverse events (not specified in detail).	Repeated intratumoral MV-CEA is safe. CEA levels correlate with viral replication. Dual-Labeling Diagnostic Assay (DLDA) may predict viral response.	[118]

oMeV: Oncolytic Measles Virus, GBM: Glioblastoma, CEA: Carcinoembryonic Antigen, MV: Measles Virus, miR: MicroRNA, JAK: Janus Kinase, PDX: Patient-Derived Xenograft, CDV: Canine Distemper Virus, scFv: Single-Chain Variable Fragment, CD46: Complement Decay-Accelerating Factor, mRNA-seq: Messenger RNA Sequencing, DLDA: Dual-Labeling Diagnostic Assay, NS: Not Specified.

**Table 6 cancers-17-02550-t006:** Preclinical and Clinical Evaluation of Oncolytic Newcastle Disease Virus (oNDV) Variants in Glioma Therapy.

oNDV Variant	Genetic Modifications/Additions	Study Model	Key Findings	Ref.
Preclinical Studies				
Wild-type NDV	None	Human GBM cell lines (GBM18, GBM27, etc.) in immunodeficient mice	IFN-I gene cluster deletion enhanced NDV replication and tumor lethality; NS1-expressing NDV overcame IFN-I resistance	[121]
NDV-2F/2HN-IFNγ	Incorporated F and HN genes from APMV-2; added human IFN-γ gene	Human PBMCs, chicken embryo fibroblasts (CEFs), Caco-2 (colon cancer) cell line	Enhanced IFN-γ expression and immune response; increased cancer cell death; evaded NDV-specific antibodies	[123]
NDV + TMZ-PLGA-NPs	Combination of wild-type NDV with Temozolomide-loaded PLGA nanoparticles	Human GBM cell lines (in vitro)	Combination therapy improved cytotoxicity over individual agents; TMZ-NPs enhanced drug stability and delivery	[124]
rNDV-PTEN (Pos. 1 vs. 2)	PTEN gene inserted between NP-P (Pos. 1) or P-M (Pos. 2) genes in NDV genome	T98G GBM cells in immunodeficient mice (in vitro and in vivo)	PTEN overexpression (especially in Pos. 1) suppressed cancer growth markers (P-Akt, hTERT); intratumoral injection more effective than intravenous delivery	[125]
**Clinical Studies**				
**Virus Used**	**Design and Intervention**	**Adverse Events**	**Notable Findings**	**Ref.**
NDV-HUJ	Phase I/II, open-label trial (n = 11); phase I: escalating weekly cycles (0.1 to 55 BIU); phase II: 3 weekly cycles of 5 days at 11 BIU, then maintenance 2x/week	No adverse events related to NDV-HUJ	NDV found in blood, urine, CSF, saliva, tumor; 1 pt had complete remission but relapsed in 3 months; anti-NDV antibodies plateaued at 8 weeks	[126]
MTH-68/H	Case series (n = 4); daily to twice daily dosing ranging from 2 × 10^7^ to 2.5 × 10^8^ PFU	Not reported	Tumor regression, neurological improvement, steroid withdrawal; long-term survival 5–9 years from diagnosis	[127]
Ulster	Nonrandomized controlled study (n = 22); ASI vaccine (NDV + cisplatin-inactivated autologous tumor) via cutaneous injection every 2 wks × 5; CG had chemo (nimustine + VM-26)	No significant adverse events with ASI	ASI induced immune response via DTH; no survival difference, but TG (ASI) had mean survival of 46 weeks vs. 48 weeks for CG	[128]
Ulster (ATV-NDV)	Nonrandomized controlled study (n = 111); NDV-inactivated autologous tumor cells injected q3–4 wks up to 8 doses post-RT	No significant adverse events with ATV-NDV	Median OS: TG 100 wks, CG-NS 49 wks, CG-S 88 wks; elevated CD8+ TILs, increased memory T cells in long-term survivors; durable immune memory	[129]

oNDV: Oncolytic Newcastle Disease Virus, GBM: Glioblastoma, IFN: Interferon, NDV: Newcastle Disease Virus, PBMCs: Peripheral Blood Mononuclear Cells, CEFs: Chicken Embryo Fibroblasts, TMZ: Temozolomide, PLGA: Poly (lactic-co-glycolic acid), NP: Nucleoprotein, P: Phosphoprotein, M: Matrix Protein, DTH: Delayed-Type Hypersensitivity, OS: Overall Survival, TILs: Tumor-Infiltrating Lymphocytes.

**Table 7 cancers-17-02550-t007:** Clinical Evaluation of Reovirus (Pelareorep) Therapy in Adult and Pediatric Gliomas: Delivery, Safety, and Immune Responses.

Virus Used	Patient Population	Design and Intervention	Adverse Events	Notable Findings	Ref.
Reovirus (Pelareorep)	Recurrent malignant gliomas	Phase I trial; intratumoral catheter-based infusion for 72 hrs	One transient grade 3 seizure; otherwise well tolerated	Virus reached tumor; histology showed cell death and immune activation; safe at high doses	[135]
Reovirus + GM-CSF	Newly diagnosed GBM	ReoGlio (2017–2020): IV pelareorep + GM-CSF during concurrent chemoradiation and adjuvant TMZ	Flu-like symptoms, reversible hypotension; no major safety concerns or treatment delays	Virus delivery to tumor confirmed; some T cell activity observed in resected tumors; early signals of immunological engagement	[136]
Reovirus	Newly diagnosed GBM	Window-of-Opportunity trial: single IV dose prior to surgery	Well tolerated	Viral RNA/proteins detected in tumor tissue; increased CD8+ T cell activity; proof of BBB crossing and immunogenic modulation	[137]
Reovirus + GM-CSF	Pediatric high-grade gliomas (incl. GBM)	Small trial (IV Pelareorep + GM-CSF)	Grade 3 hyponatremia (1 patient); overall well tolerated	All patients progressed within weeks; early antibody neutralization likely impacted efficacy; confirms tumor entry via IV but poor clinical outcome	[138]

GBM: Glioblastoma, IV: Intravenous, TMZ: Temozolomide, GM-CSF: Granulocyte-Macrophage Colony-Stimulating Factor, BBB: Blood–Brain Barrier, CD8+: Cytotoxic T-Cells, TILs: Tumor-Infiltrating Lymphocytes, RNA: Ribonucleic Acid, PFU: Plaque-Forming Units.

**Table 8 cancers-17-02550-t008:** Preclinical and Clinical Evaluation of Genetically Modified Oncolytic Retroviruses in Glioblastoma Therapy.

Virus Variant	Genetic Modifications/Additions	Study Model	Key Findings	Ref.
Pre-clinical Studies				
MLV-based RCR	RCR vector expressing yeast cytosine deaminase gene with 5-FC prodrug	Cell lines, mouse models	Single injection led to widespread tumor infection and enhanced survival; tumor-selective expression of suicide gene.	[139]
Toca 511	Amphotropic MLV encoding modified yeast cytosine deaminase 2 (yCD2)	In vitro, in vivo	Tumor-targeted delivery; converts 5-FC to 5-FU; safe replication; efficient and stable gene delivery in glioblastoma.	[140]
oFV-GFP	oFV vector with chimpanzee virus PAN1/PAN2 genes and green fluorescent protein (GFP) transgene	GBM cells, mouse models	Safe, stable, and tumor-persistent; spreads well even in slow-growing tumors; prolongs survival; can reactivate after dormancy.	[141]
oFV-GFP, oFV-TK, oFV-iCasp9	GFP, thymidine kinase (TK), and inducible caspase 9 (iCasp9) inserted into oFV vectors	Human GBM cells, mouse models	Effective long-term gene expression; GFP expression persisted for 66 days; larger genes like iCasp9 lost over time during replication in tumors.	[142]
**Clinical Studies**				
**Virus Used**	**Design and Intervention**	**Adverse Events**	**Notable Findings**	**Ref.**
Toca 511	45 patients with glioblastoma	Phase 1 trial; virus injected after surgery followed by oral Toca FC and standard chemoradiotherapy	Grade 3 asthenia, hydrocephalus, thrombocytopenia, procedural pain	[143]
Toca 511	56 patients with recurrent high-grade glioma	Phase 1 dose-escalation; intratumoral virus injection post-surgery + Toca FC oral prodrug	Grade 2 skin rash, mucositis, facial swelling; Grade 3 hemorrhagic enteritis and colitis	[144]
Toca 511	56 high-grade glioma patients	Phase 1; integrated omics analysis (WES, RNA-seq, ELISA) on tumor and blood to assess response	Not disclosed	[145]
Toca 511	127 high-grade glioma patients	Large study using extended-release Toca FC and molecular monitoring (RNA/DNA tracking, integration site analysis, mutation profiling)	Hematologic adverse events including lymphoma	[146]

## Data Availability

The datasets generated during and/or analyzed during the current study are available from the corresponding author on reasonable request.

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
