# Peer review of "Oncolytic Therapies for Glioblastoma: Advances, Challenges, and Future Perspectives"

_cancers, 2025, doi:10.3390/cancers17152550_

Round 1

Reviewer 1 Report

Comments and Suggestions for Authors

The manuscript "Oncolytic Therapies for Glioblastoma: Advances, Challenges, and Future Perspectives" is a yet another review on the progress of oncolytic virotherapy and bacteriotherapy against one the deadliest malignancies. It is well-organized in a classical manner, the references seem to be up-to-date, tabular data are explicit, therefore it is possible to publish it so that it may serve as a good milestone in the understanding of perspectives to find a decisive cure for glioblastoma (GBM). However, the paper needs an overhaul, since anti-plagiarism software indicate almost 17% matches with already published texts. Upon examining these matches in detail, I had an impression that all of them are either merely technical or rather short to suspect copy-pasting, therefore, I think that these should not be viewed upon as plagiarism, however, it is necessary to fix all of them, thus publication after a major revision is the best option. 

Minor:

  1. In the introduction it should be mentioned that anti-VEGF treatments (bevacizumab particularly) also provide significant benefit for some patients. 

  1. It is not clear how bacteria are allies in the fight against GBM - this is nice to include bacteriotherapy together with oncolytic viruses but any practical perspectives for bacterial injections into the brain are meager to say the least...

  1. Figures look like prepared using Biorender.

  1. The role of DAMPS and interferon pathway are not fully described.

  1. There is a logical jump from "HSV-1 is particularly advantageous due to its extensive characterization as a human pathogen" to the actual use, in fact it was a huge work to tame it so that to minimize pathogenicity and become suitable as an OV.

  1. Newcastle virus is highly dangerous for poultry, therefore, it clinical use is doubtful.

  1. "Oncolytic peptides" should be deleted since this subject deserves a separate review due to the vast amount of information on peptides and proteins capable of such effects. 

  1. Polio virus-based OVs are not sufficiently described. Also, some researchers tested Zika etc preclincally.

  1. Retroviruses seem largely forgotten, meanwhile, there were interesting reports of successes and failures.

  1. The problem of systemic delivery should be discussed in more detail.

Author Response

Reviewer 1 (all related edits are highlighted with yellow in the manuscript)  

The manuscript "Oncolytic Therapies for Glioblastoma: Advances, Challenges, and Future Perspectives" is a yet another review on the progress of oncolytic virotherapy and bacteriotherapy against one the deadliest malignancies. It is well-organized in a classical manner, the references seem to be up-to-date, tabular data are explicit, therefore it is possible to publish it so that it may serve as a good milestone in the understanding of perspectives to find a decisive cure for glioblastoma (GBM). However, the paper needs an overhaul, since anti-plagiarism software indicate almost 17% matches with already published texts. Upon examining these matches in detail, I had an impression that all of them are either merely technical or rather short to suspect copy-pasting, therefore, I think that these should not be viewed upon as plagiarism, however, it is necessary to fix all of them, thus publication after a major revision is the best option.

REPLY: We appreciate your comment regarding this paper. The whole revised manuscript has been checked based on your comments and the plagiarism rate has been lowered as much as possible and now it is %8 and all are as you mentioned merely technical or rather short to suspect copy-pasting. The Turnitin report also is uploaded as an attachment.

Minor:

  1. In the introduction it should be mentioned that anti-VEGF treatments (bevacizumab particularly) also provide significant benefit for some patients.

REPLY: We appreciate the reviewer’s insightful suggestion.  We have added a paragraph about the anti-VEGF treatments in the introduction section as suggested (Page: 2; Lines: 46-59).

  1. It is not clear how bacteria are allies in the fight against GBM - this is nice to include bacteriotherapy together with oncolytic viruses but any practical perspectives for bacterial injections into the brain are meager to say the least...

REPLY: Thank you so much for your kind and valuable comment. Up to date few studies have been investigated the bacteriotherapy applications in GBM, so actually there is no that much of data regarding this. However, we have added all the reported possible articles and mechanisms about how bacteria allies in the fight against GBM at the end of the bacteriotherapy section (Page: 23-24; Lines: 668-688).

  1. Figures look like prepared using Biorender.

 REPLY: Thank you so much for comment. Yes, all the figures have been prepared using Biorender. According to our account plan we have added the sentence “Created with BioRender.com” below each figure (Pages: 4,5,25).

  1. The role of DAMPS and interferon pathway are not fully described.

REPLY: We appreciate the reviewer’s insightful suggestion. We have added a paragraph about the DAMPS and interferon pathway to the introduction section (Page: 3; Lines: 93-104).

  1. There is a logical jump from "HSV-1 is particularly advantageous due to its extensive characterization as a human pathogen" to the actual use, in fact it was a huge work to tame it so that to minimize pathogenicity and become suitable as an OV.

 REPLY: We appreciate the reviewer’s insightful comment on this point which we do agree with. In response we have revised the sentence based on your comment to avoid and misleading information especially about the safety of HSV-1 and we do believe it is better now (Page: 5; Lines: 168-172).

  1. Newcastle virus is highly dangerous for poultry, therefore, it clinical use is doubtful.

 REPLY: Thank you so much for drawing our attention toward this point, we appreciate your insightful comment. You are alright NDV is a highly contagious and lethal avian virus, so its use in humans might raise regulatory, ecological, and zoonotic safety concerns. However, as demonstrated in multiple preclinical and clinical studies (summarized in Table 6) NDV has shown a favorable safety profile in humans, with no serious adverse events reported. Its clinical application also has not been limited to GBM; NDV-based oncolytic virotherapy has also been explored in various solid tumors, including melanoma, renal cell carcinoma, and ovarian cancer. These studies collectively support the feasibility of NDV use under controlled conditions and with appropriate safety protocols.  Thank you again for raising this important concern. We have addressed this issue by adding a dedicated paragraph in the manuscript that discusses the safety considerations of NDV, including the concern regarding its pathogenicity, and the strategies to mitigate associated risks in clinical use. We believe this addition strengthens the discussion on NDV’s translational potential and clinical applicability (Page: 16; Lines: 391-405).

  1. "Oncolytic peptides" should be deleted since this subject deserves a separate review due to the vast amount of information on peptides and proteins capable of such effects. 

 REPLY: We appreciate your insightful suggestion about deleting the "Oncolytic peptides" section. While we acknowledge that this topic is indeed broad and could warrant a separate, dedicated review, we believe it is still appropriate to retain a brief discussion within our manuscript. Given that this review is part of a special issue focused on recent advancements in cancer therapies, mentioning oncolytic peptides helps ensure that the scope and title of the paper accurately reflect the breadth of current therapeutic strategies. We also note that this is not a systematic review, so it is not necessary to include all available data in exhaustive detail. Our intention is to provide a comprehensive yet concise overview. However, if you have specific concerns or suggestions about how this section could be improved or streamlined, we would be happy to revise accordingly.

  1. Polio virus-based OVs are not sufficiently described. Also, some researchers tested Zika etc preclinically.

 REPLY: We appreciate the reviewer’s insightful suggestion. A paragraph about the treatment applications of Polio virus (Page: 20; Lines: 507-523) and Zika Virus (Page: 20; Lines: 507-523) in GBM has been added to the manuscript (Page: 20-21; Lines: 524-539).

  1. Retroviruses seem largely forgotten, meanwhile, there were interesting reports of successes and failures.

 REPLY: We appreciate the reviewer’s insightful suggestion. A paragraph about the treatment applications of Retroviruses in GBM (Page: 18,19; Lines: 447-464).

  1. The problem of systemic delivery should be discussed in more detail.

REPLY: Thank you so much for your valuable comment. We have added a more detailed paragraph about the problem of systemic delivery and other possible challenges (Page: 27; Lines: 805-832).

We appreciate your continued support and look forward to your feedback and last decision on the revised manuscript.

Reviewer 2 Report

Comments and Suggestions for Authors

Review of Manuscript “Oncolytic Therapies for Glioblastoma: Advances, Challenges, and Future Perspectives” byOmar Alomari et al..

Different oncolytic viruses (OVs) have emerged as promising modalities for the treatment of glioblastomas (GBM), especially in combination with more conventional treatments such as chemotherapy, radiation or also in combination with immune checkpoint inhibitors. In the present manuscript the authors review the properties of various kinds of genetically modified OVs such as herpes simplex virus (HSV), adenovirus (AdV), measles virus (MeV), poliovirus (PV), Newcastle disease virus (NDV), reovirus and vaccinia virus (VV) and their application in preclinical and clinical studies of GBM. In addition, treatment regimens using oncolytic bacteria or peptides are reviewed. The review provides a good compilation of the available literature and the principal properties of the different OVs are nicely described. However, in view of the large number of very recent reviews on this topic, mostly cited also by the authors, one must raise the fundamental question of the actual added value of yet another review. Especially in the sections on oncolytic HSV and oncolytic adenoviruses the reader is left with a compilation of the available studies in form of tables and a corresponding short summary. For readers, who already have a strong background in OVs, this may be an adequate form. To my opinion, however, it would have been a better strategy to select the approaches that the authors consider particularly interesting and/or promising and to explain these in the text in a way that is also understandable for readers who are not quite so familiar with the topic. This would also offer the opportunity to describe the mode of action of the molecular players involved such as immunomodulatory genes and immune checkpoints with their corresponding inhibitors. The tables may be left as additional information. More individual issues to be addressed are listed in detail below.      

Major points:

1) Line numbers were missing in the manuscript, which makes writing the review unnecessarily difficult.

2) Page 14, section 2.3: “However, antiviral resistance mechanisms in tumor cells remain a barrier. In one recent study, MeV encoding miR-122 reduced target protein levels by 40%, though Drosha-dependent nuclear retention limited miRNA production [111]. Another study using patient-derived xenograft (PDX) models identified the JAK1 pathway as a key regulator of MeV replication, and combining MeV with the JAK1 inhibitor ruxolitinib improved viral replication and treatment efficacy, highlighting the need for further integration with immune-modulating strategies such as T-cell monitoring [112].”

Wrong assignment of references, these should be references no. 110 and 111, respectively, as also stated correctly in table 5. Furthermore, Reference no. 110 describing a MeV encoding miR-122 only addresses the principal possibility of expression of miRNA’s by MeV, but falls short of targeting any host genes involved in antiviral resistance mechanisms.

3) References no. 147 to 160 are not correctly assigned (the reference referred to in the text as no. 146 seems to be missing and all the subsequent references are therefore shifted in the list by one number. This is resolved with reference no. 160, which is quoted two times in the text, the first quote is incorrect, the second one is correct, which reestablishes the correct numbering).

4) Parvovirus H1 is also a possible viral treatment modality for GBM and should be included in the review.   

Minor points:

1) Page 4, section 2.1: “These viruses can establish lifelong latent infections, particularly in neural tissues, and may reactivate under conditions such as immunosuppression [15]. Furthermore, under certain conditions such as immunosuppression, they may reactivate from latency and cause lytic infections, which contributes to their clinical significance“.

The two sentences are highly redundant and should be reformulated.

2) Page 5, section 2.1: “HSV-1 inhibits protein synthesis by preventing PKR activation through its ICP 34.5 protein,”

Since PKR activation reduces protein synthesis, the prevention of PKR activation should actually lead to an increase in protein synthesis.

3) Page 12, section 2.2.2: “..as a therapeutic GBM modality, a malignancy known for its aggressive nature and resistance to conventional therapies..”

This has already been described in detail in the introduction section.

4) Section 2.4 Newcastle Disease Virus: Reference no. 105 is not correctly assigned. Should probably read no. 115.

Author Response

Reviewer 2 (all related edits are highlighted with Green in the manuscript)    

Different oncolytic viruses (OVs) have emerged as promising modalities for the treatment of glioblastomas (GBM), especially in combination with more conventional treatments such as chemotherapy, radiation or also in combination with immune checkpoint inhibitors. In the present manuscript the authors review the properties of various kinds of genetically modified OVs such as herpes simplex virus (HSV), adenovirus (AdV), measles virus (MeV), poliovirus (PV), Newcastle disease virus (NDV), reovirus and vaccinia virus (VV) and their application in preclinical and clinical studies of GBM. In addition, treatment regimens using oncolytic bacteria or peptides are reviewed. The review provides a good compilation of the available literature and the principal properties of the different OVs are nicely described. However, in view of the large number of very recent reviews on this topic, mostly cited also by the authors, one must raise the fundamental question of the actual added value of yet another review. Especially in the sections on oncolytic HSV and oncolytic adenoviruses the reader is left with a compilation of the available studies in form of tables and a corresponding short summary. For readers, who already have a strong background in OVs, this may be an adequate form. To my opinion, however, it would have been a better strategy to select the approaches that the authors consider particularly interesting and/or promising and to explain these in the text in a way that is also understandable for readers who are not quite so familiar with the topic. This would also offer the opportunity to describe the mode of action of the molecular players involved such as immunomodulatory genes and immune checkpoints with their corresponding inhibitors. The tables may be left as additional information. More individual issues to be addressed are listed in detail below.

REPLY:  We sincerely thank the reviewer for their thoughtful feedback and valuable suggestions. We agree that the field of oncolytic virotherapy has been the focus of several recent reviews; however, we believe the added value of our work lies in its broad and integrative scope. This review uniquely compiles and contrasts the major classes of oncolytic agents including viruses, bacteria, and peptides, while also exploring combination strategies with existing GBM treatments and outlining the current challenges and future directions. We aimed to provide a comprehensive yet accessible resource for both seasoned researchers and newcomers to the field. Regarding the presentation format, we acknowledge the limitations of summarizing complex mechanistic details and the targeted pathophysiological pathways for each virus within a single manuscript, especially in the context of a special issue focused on recent therapeutic advances. Our goal was to provide a concise overview of each oncolytic strategy, allowing readers to grasp the landscape quickly, while directing those interested in deeper mechanistic insights to more specialized reviews. Nonetheless, we appreciate your recommendation and will consider any future comments to enhance the current data presenting strategy and ideas flow in the paper.

  • Major points:

  • Line numbers were missing in the manuscript, which makes writing the review unnecessarily difficult.

REPLY: Thank you for your feedback. We apologize for the oversight regarding the missing line numbers in the manuscript. We have now revised the document and included line numbers throughout to facilitate easier review. Please let us know if any additional formatting changes are required.

  • Page 14, section 2.3: “However, antiviral resistance mechanisms in tumor cells remain a barrier. In one recent study, MeV encoding miR-122 reduced target protein levels by 40%, though Drosha-dependent nuclear retention limited miRNA production [111]. Another study using patient-derived xenograft (PDX) models identified the JAK1 pathway as a key regulator of MeV replication, and combining MeV with the JAK1 inhibitor ruxolitinib improved viral replication and treatment efficacy, highlighting the need for further integration with immune-modulating strategies such as T-cell monitoring [112].” Wrong assignment of references, these should be references no. 110 and 111, respectively, as also stated correctly in table 5. Furthermore, Reference no. 110 describing a MeV encoding miR-122 only addresses the principal possibility of expression of miRNA’s by MeV, but falls short of targeting any host genes involved in antiviral resistance mechanisms.

REPLY: We appreciate the reviewer’s valuable comment regarding such important points. The references have been reassigned accordingly. Data related to Reference no. 110 (116 currently) also has been rewritten as a principal possibility results of the study (Page: 14; Lines: 351-358).

  • References no. 147 to 160 are not correctly assigned (the reference referred to in the text as no. 146 seems to be missing and all the subsequent references are therefore shifted in the list by one number. This is resolved with reference no. 160, which is quoted two times in the text, the first quote is incorrect, the second one is correct, which reestablishes the correct numbering).

REPLY: Thank you for your comment. Both references have been reassigned accordingly and the whole reference list has been checked again (Page: 14; Lines: 351-358).

  • Parvovirus H1 is also a possible viral treatment modality for GBM and should be included in the review.   

 REPLY: We appreciate the reviewer’s insightful suggestion. A paragraph about the treatment applications of Parvovirus H1 in GBM has been added to the manuscript (Page: 20; Lines: 487-506).

  • Minor points:

1) Page 4, section 2.1: “These viruses can establish lifelong latent infections, particularly in neural tissues, and may reactivate under conditions such as immunosuppression [15]. Furthermore, under certain conditions such as immunosuppression, they may reactivate from latency and cause lytic infections, which contributes to their clinical significance“. The two sentences are highly redundant and should be reformulated.

 REPLY: Thank you for pointing out the redundancy in these sentences. We have revised the text to improve clarity and avoid repetition (Page: 5; Lines: 154-157).

2) Page 5, section 2.1: “HSV-1 inhibits protein synthesis by preventing PKR activation through its ICP 34.5 protein,” Since PKR activation reduces protein synthesis, the prevention of PKR activation should actually lead to an increase in protein synthesis.

REPLY: Thank you for this important observation. You are absolutely correct; our original wording was inaccurate. We have revised the sentence to clarify that HSV-1 promotes protein synthesis by counteracting PKR-mediated inhibition. The corrected sentence now reads: “HSV-1 preserves protein synthesis during infection by preventing the PKR-mediated phosphorylation of eIF2α through its ICP34.5 protein, thereby counteracting the host's antiviral response, …..” .

We appreciate your attention to this detail and have updated the manuscript accordingly (Page: 5; Lines: 187-189).

3) Page 12, section 2.2.2: “..as a therapeutic GBM modality, a malignancy known for its aggressive nature and resistance to conventional therapies..” This has already been described in detail in the introduction section.

REPLY: Thank you for this important observation. The sentence has been removed as suggested.

4) Section 2.4 Newcastle Disease Virus: Reference no. 105 is not correctly assigned. Should probably read no. 115.

REPLY: Thank you for this important observation. You are alright it should be 115 instead of 116, the reference has been changed to 115 (currently 121).

We appreciate your continued support and look forward to your feedback and last decision on the revised manuscript.

Round 2

Reviewer 1 Report

Comments and Suggestions for Authors

All concerns have been adressed with exceedingly careful attention, therefore, it is a pleasure to recommend the revised manuscript for publication in Cancers.

Author Response

All concerns have been adressed with exceedingly careful attention, therefore, it is a pleasure to recommend the revised manuscript for publication in Cancers.

REPLY: We sincerely appreciate your thoughtful evaluation and kind recommendation. We are grateful for your insightful feedback throughout the review process, which greatly contributed to the improvement of our manuscript. Thank you again for your time and support.

Reviewer 2 Report

Comments and Suggestions for Authors

Review of revised version of manuscript “Oncolytic Therapies for Glioblastoma: Advances, Challenges, and Future Perspectives” by Omar Alomari et al..

In the revised version, the authors have addressed the detailed points raised in my review of the original manuscript in a detailed and satisfactory fashion. I still have some reservations regarding the suitability of the review for newcomers in the field, due to the limited explanations of the complex mechanisms involved in the selective killing of tumor cells by oncolytic viruses. However, I can go with the argumentation of the authors that their goal was to provide a concise overview directing those interested in the mechanistic details to more specialized reviews or original papers from the truly comprehensive reference list.  

Minor point:

1) Page 20, line 488: Incomplete sentence, should probably read “to normal cells“ at the end of the sentence.

Author Response

In the revised version, the authors have addressed the detailed points raised in my review of the original manuscript in a detailed and satisfactory fashion. I still have some reservations regarding the suitability of the review for newcomers in the field, due to the limited explanations of the complex mechanisms involved in the selective killing of tumor cells by oncolytic viruses. However, I can go with the argumentation of the authors that their goal was to provide a concise overview directing those interested in the mechanistic details to more specialized reviews or original papers from the truly comprehensive reference list. 

REPLY:  We thank the reviewer for the positive and constructive feedback, as well as for acknowledging our efforts to address the detailed points raised in the initial review. We also appreciate the reviewer’s thoughtful consideration of our rationale regarding the scope of the manuscript. We are grateful for the reviewer’s understanding and support.

Minor point:

1) Page 20, line 488: Incomplete sentence, should probably read “to normal cells“ at the end of the sentence.. 

REPLY:  We thank the reviewer for pointing out this oversight. We have corrected the incomplete sentence on page 20, line 488 as suggested.